# Single trial neuronal activity dynamics of attentional intensity in monkey visual area V4

Supriya Ghosh [1✉] & John H. R. Maunsell [1]

Understanding how activity of visual neurons represents distinct components of attention and their dynamics that account for improved visual performance remains elusive because single-unit experiments have not isolated the intensive aspect of attention from attentional selectivity. We isolated attentional intensity and its single trial dynamics as determined by spatially non-selective attentional performance in an orientation discrimination task while recording from neurons in monkey visual area V4. We found that attentional intensity is a distinct cognitive signal that can be distinguished from spatial selectivity, reward expectations and motor actions. V4 spiking on single trials encodes a combination of sensory and cognitive signals on different time scales. Attentional intensity and the detection of behaviorally relevant sensory signals are well represented, but immediate reward expectation and behavioral choices are poorly represented in V4 spiking. These results provide a detailed representation of perceptual and cognitive signals in V4 that are crucial for attentional performance.

[1] Department of Neurobiology and Neuroscience Institute, The University of Chicago, Chicago, IL, USA. ✉email: sghosh5@uchicago.edu

Selective attention greatly improves performance by enhancing the processing of the fraction of available sensory information that is most behaviorally relevant. Neuronal responses across many cortical and subcortical visual areas in the brain are strongly modulated when animals covertly shift their focus of attention in visual detection tasks[1–4]. When a monkey's attention is directed toward the location of a neuron's receptive field (RF), improvement in perceptual performance in that region is accompanied by greater spike rates[5,6], reduced individual response variance and decreased pairwise spike-count correlations[7,8]. Many psychological and neurophysiological studies have proposed that attention might consist of multiple distinct neurobiological mechanisms[9–11]. Recent studies have identified two behavioral components of attentional performance, behavioral sensitivity (d′) and response criterion, that are differentially represented in different visual structures. Neuronal responses in visual cortical area V4 are selectively modulated by changes in behavioral sensitivity (d′) and remain unaffected by the changes in response criterion[6]. Unlike V4, neurons in the lateral prefrontal cortex are strongly modulated by either changes in response criterion or sensitivity[12].

Although attention commonly refers to selective focus, it also has another fundamental aspect—amplitude or intensity[13], which is separable from selectivity (Fig. 1). For instance, when attention is shared between two spatial locations, the directed attention can be more or less intense with the same selectivity (radial points, Fig. 1b). Similarly, for the same intensity, attention can be highly selective or nonselective (moving from off diagonal toward the diagonal along an arc, Fig. 1b). Schematically, attentional intensity is represented here as an objective function of resultant performance across targets. It may not fully capture all aspects of perceived subjective mental effort[13]. Attention and arousal are thought to be closely related. An individual is expected to be aroused in order to maintain a high level of performance while engaged in a demanding task. It is plausible that the intensive component of attention is a specific type of more general arousal, but it remains unknown how this top-down intensity signal is represented by cortical neuronal activity. In most previous studies, neuronal modulation in V4 attributed to visual attention was limited to the changes in attentional selectivity. In response to increased task demand, V4 neuronal activity is enhanced as a result of higher cognitive engagement[14,15], which is thought to be closely associated with the intensive aspect of attention[11]. It has also been proposed that spike rates of V4 neurons can signal absolute reward size[16] and motor action[17,18] associated with a visual target inside their RFs independent of attentional performance. Given the strong experimental evidence for coexistence of neuronal correlates of distinct cognitive and behaviorally relevant information in V4, the exact contribution of these covariates and their dynamics on spiking activities remains unknown.

An interesting hypothesis is that representations of fundamental aspects of attention and task-relevant variables in V4 might be multiplexed with distinct dynamics. In the context of attention-demanding goal-directed tasks, isolating neuronal substrates of attentional intensity from attentional selectivity and any other relevant sensory or task variables is crucial to better understand how visual cortex precisely contributes to distinct components of attention. We manipulated the level of monkeys' attentional intensity using a novel spatial attention task in which their overall attentional intensity was switched between high and low levels with no change in attentional selectivity for stimuli at one location over another, over small block of trials (green markers, Fig. 1b). Dashed arcs around the origin mark isopleths of constant total attentional intensity or iso-intensity across the two stimulus locations. Movements perpendicular to these lines correspond to changes in attentional intensity; movements along these lines correspond to changes in attentional selectivity. Here we have considered an objective definition of attentional intensity as measured by spatially nonselective behavioral performance (d′) (Fig. 1b). Using simultaneous electrophysiological recordings from a population of V4 neurons and computational models, we found that attentional intensity modulates V4 neurons independent of spatial or featural attention. Additionally, the relative reward size for the hits and correct rejections (CRs) were varied within blocks in an uncued way in order to encourage the animals to maintain a behavioral criterion near 0. This variance in reward size allowed us to produce single-trial estimates of the dynamics of neuronal correlates of reward expectation as well as attentional intensity. Using a generalized linear model (GLM) on single-trial spike counts of individual V4 neurons, we quantitatively described contributions of temporally overlapping neuronal response components associated with attentional intensity, reward expectation, sensory information, and perceptual detection. This approach revealed both steady state and time varying multiplexed cognitive and task-relevant signals encoded in V4, enabled the assessment of a broader spectrum of cognitive functions to which visual cortical areas can contribute dynamically.

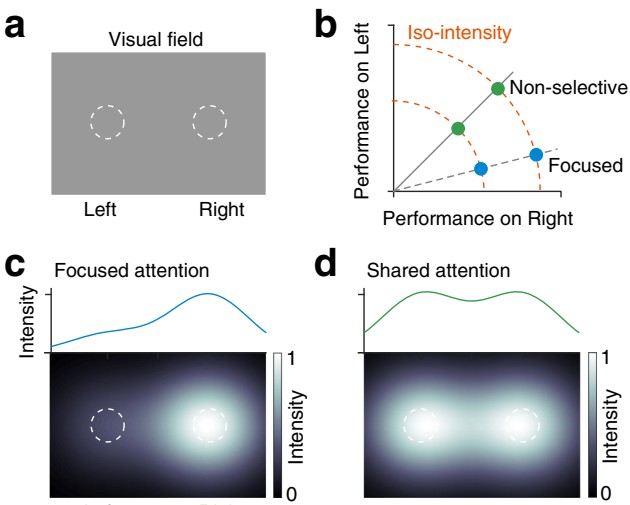

**Fig. 1 Attentional components, selectivity and intensity. a** Visual field. Dashed circles, locations of visual spatial attention. **b** Attention operating characteristic (AOC). The spatial distribution of performance, as visual attention is being shared between the two locations, left and right. Green circles, attention is equally shared between the left and the right location with no selectivity. Blue circles, selective attention is directed toward the right location. Dashed arcs, path of constant intensity of attention (iso-intensity). **c** Bottom, simulated spatial distribution of attentional intensity when attention is focused in right. Top, average attentional intensity along horizontal. **d** Same as in (**c**), for equally shared attention between left and right.

## Results

**Behavioral control of attentional intensity**. We trained two rhesus monkeys to distribute their visual spatial attention between the left and right hemifields while doing an orientation detection task (Fig. 2a). The animal held its gaze on a central fixation spot throughout each trial. After a randomly varying period of fixation, two Gabor sample stimuli appeared briefly for 200 ms. This was followed by a brief delay (200–300 ms), and then a single Gabor test stimulus appeared at one of the two sample locations (selected pseudo-randomly). If the orientation of the test stimulus differed from that of the sample stimulus that

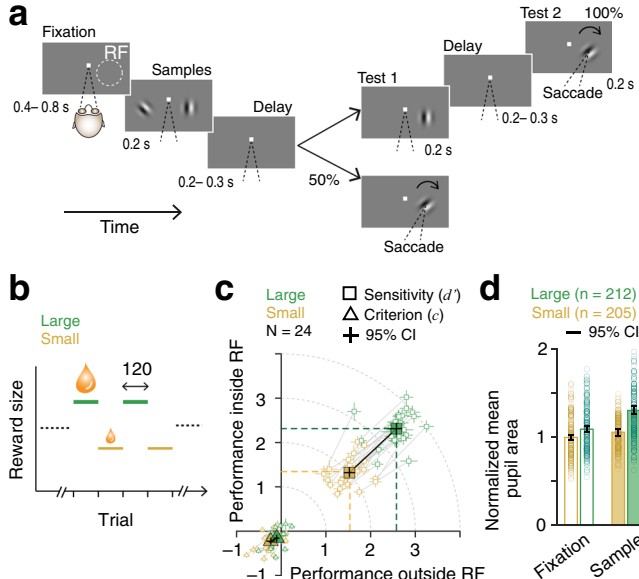

**Fig. 2 Manipulation of attentional intensity using differential reward outcomes. a** Visual spatial attention task. Monkeys were required to fixate, attend to sample stimuli (Gabors) presented in both hemifields (inside and outside of recorded neurons' receptive field (RF)), and report an orientation change that occurred in one of the two test intervals by making a saccade to the stimulus location. **b** Unsignaled change in reward size between large and small values over blocks of 120 trials. **c** Attention operating characteristic (AOC) curve, indicating behavioral sensitivity (d', circles) and criterion (c, triangles) on individual sessions and their average (solid markers) for test stimuli inside and outside RF during two reward conditions, large and small (24 Total sessions: 9 monkey P; 15 monkey S). Dashed colored lines indicate average d' in each hemifield. Lines connect two reward conditions within a session. Data are presented as mean with 95% confidence intervals. **d** Block-averaged pupil area during pre-stimulus fixation and sample stimulus periods. Pupil area was normalized to mean of pre-stimulus fixation during small-reward trial blocks. Data are presented as mean with 95% confidence intervals.

had appeared in that location (a target), the monkey had to rapidly make a saccade to the stimulus to earn a juice reward. On a random half of the trials, the orientation of test stimulus was unchanged (a nontarget) and the monkey was required to maintain fixation. In that case, a second test stimulus that always had a different orientation was presented after a short delay and monkeys needed to saccade to this stimulus to earn a reward. Reward values for correct responses were always the same on both sides and fixed, either small or large, within alternating blocks of 120 trials. Stimulus parameters remained unchanged throughout all trial blocks within a session, except for the reward size. The transitions between blocks were unsignaled and sequential (small to large or large to small, Fig. 2b). The monkey therefore had to estimate reward expectancy based on previous trials to adjust its behavioral strategy in allocating attention between blocks.

Varying rewards in different blocks of trials allowed us to behaviorally control the amount of attention the animal directed to each stimulus location. Animals were encouraged to maintain a behavioral target/nontarget bias (criterion, c) close to 0 by providing rewards of different sizes for hits and CRs (Supplementary Fig. 1). Because the reward sizes for targets were always the same in both hemifields, animals typically allocated equal attention to both sides within either the high- or low-reward condition. Figure 2c plots behavioral sensitivity (d') and response criterion (c) for behavioral performance on the first test stimuli

on the left and right side, for both the large- and small-reward conditions. Solid lines join average sensitivities from alternating high- and low-reward blocks from a single day. Sensitivities on both locations are well balanced (no net attentional selectivity) during most individual sessions. Larger rewards strongly motivated the animals to increase their attentional intensity compared to small-reward trial blocks. A fourfold increase in reward size (median small reward 131 μl, quartiles 108 and 150 μl; median large reward 522 μl, quartiles 469 and 570 μl) increased overall behavioral d' (see "Methods") by 71% (mean small 2.03, SEM 0.07; mean large 3.48, SEM 0.08, $p = 10^{-17}$, paired t-test; Fig. 2c and Supplementary Table 1) and proportion of correct responses (Supplementary Fig. 2 and Supplementary Table 2). Behavioral response criterion remained near 0 regardless of reward size (Fig. 2c and Supplementary Table 1).

Non-luminance mediated task-evoked increases of pupil size are commonly considered a proxy for arousal, attentional engagement and are sensitive to task demands across species[19,20] (for review[21]). Throughout the task, animals maintained a stable fixation during which pupil area remains elevated for large-reward trials relative to small-reward trials (Fig. 2d and Supplementary Fig. 3). Large reward increased pupil area during fixation by 9% (all values normalized to fixation on small-reward trials, mean large 1.09, SEM 0.02) and during sample stimuli by 24% (mean small 1.05, SEM 0.02, mean large 1.31, SEM 0.02) (repeated measures ANOVA, effect of reward size, $F_{(1, 415)} = 45.5$, $p = 10^{-10}$; effect of sample stimulus, $F_{(1, 415)} = 180.12$, $p = 10^{-15}$; Fig. 2d). While the within block trial-averaged behavioral d' was strongly positively correlated with both reward size and pupil area during sample stimuli, pupil area failed to exhibit any correlation with reward size (partial correlation, reward $- d'$, $\rho = 0.68$, $p = 10^{-55}$; pupil area $- d'$, $\rho = 0.23$, $p = 10^{-5}$; reward $-$ pupil area, $\rho = 0.08$; $p = 0.09$; $n = 417$ blocks of trials; Supplementary Fig. 4). Increased pupil size is often taken as a signature of an increase in a subject's overall level of arousal. While it is conceivable that attentional intensity as defined in our task is identical to overall arousal, it might not be. For example, high attentional intensity in our task might correspond to greater attention to sensory rather than cognitive signals, or greater attention to visual rather than auditory signals (see "Discussion"). While the precise relationship between attentional intensity and overall arousal remains to be determined, the data from Fig. 2d show that at least some component of the pupil size modulation is associated with the visual attentional intensity controlled by our task.

### Responses of V4 neurons increase with increasing attentional intensity

Previous studies have shown that attention-related modulation of V4 neuronal responses increases when tasks demand more cognitive engagement[14,15]. However, in those studies, increases in task difficulty always covaried with increases in behavioral selectivity for one visual field location over another. A more recent study similarly showed that the V4 neurons respond more strongly when monkeys shift their attention to increase their behavioral d' at the recorded neurons' RF location relative to another location in the opposite hemifield[6], a manipulation that also covaried selectivity and sensitivity. Our task design allowed us to change attentional intensity with no appreciable change in the selectivity for one spatial location over the other (Fig. 2c).

We recorded from total 970 single units and small multiunit clusters (single unit, 298; multiunit, 672) during 24 recording sessions from the two monkeys (monkey P, 9 sessions, 407 units; monkey S, 15 sessions, 563 units) using 96 channel multielectrode arrays chronically implanted in V4 in the superficial prelunate gyrus of one hemisphere (Supplementary Fig. 5). Trial-averaged

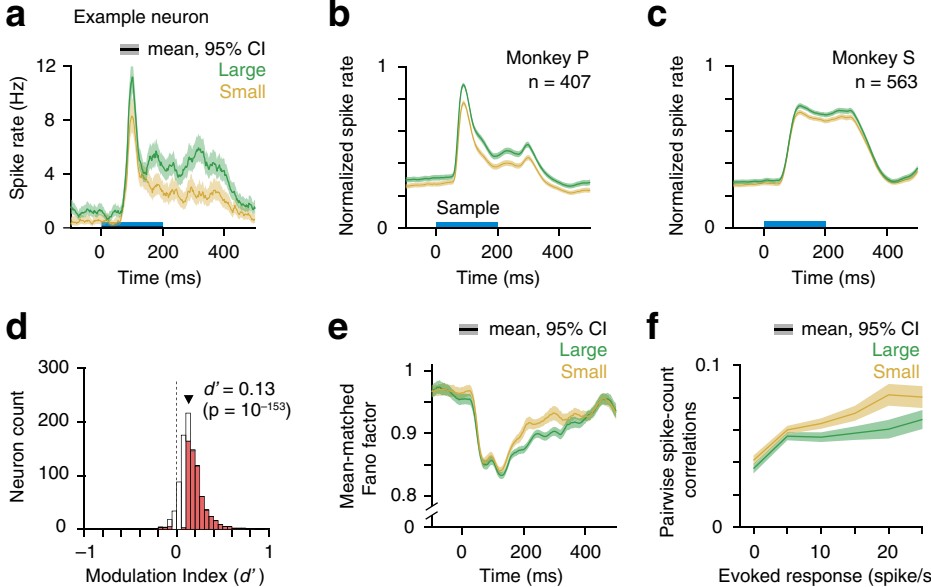

**Fig. 3 Higher attentional intensity increased spiking of neurons in area V4. a** Peri-stimulus time histograms (PSTH) of spike rates of correct trials in large- and small-reward blocks for an example neuron in V4. Single-trial spike counts were binned at 2 ms, smoothed with $\sigma = 15$ ms half Gaussian and then aligned at the onset of sample stimulus. Horizontal bar, sample stimulus presentation. Population PSTHs for monkey P (**b**) and monkey S (**c**). For population average, spike rates of each neuron were normalized to its peak response within 60–260 ms from sample stimulus onset (monkey P, $n = 407$; monkey S, $n = 563$). Error bars, 95% confidence intervals (bootstrap, $n = 10^4$). **d** Distribution of neuronal modulation indices (MI) of all units from both monkey P and S ($n = 970$). MI was measured by neuronal $d'$ (see "Methods"). Red bars, neurons with MI values significantly different from 0 ($n = 595/970$, $p < 0.05$; two-sided $t$-test). White bars, nonsignificant MI. Solid triangle, population MI averaged across all units ($n = 970$, $p = 10^{-153}$, two-sided $t$-test). **e** Mean-matched Fano factor. Error bars, 95% confidence intervals. **f** Pairwise correlations between spike counts of simultaneously recorded neurons over 60–260 ms from sample onset ($n = 20,005$ pairs, all units) and binned according to their evoked responses (geometric mean). Error bars, 95% confidence intervals.

population spike rates during the pre-sample fixation period for correctly completed trials (hits and CRs) did not differ between the two reward sizes ($p = 0.06$; Supplementary Fig. 6). Spike counts during the sample stimuli period increased in large-reward trial blocks when the monkeys' attentional intensity was higher (Fig. 3a–c). Average responses were low because the stimulus was patently suboptimal for most of the simultaneously recorded neurons. The offset response latency was longer than the onset latency. A similar offset latency difference in V4 and IT neurons was found in several previous studies[6,22,23]. A strong top-down modulation and activation of recurrent network could increase the offset latency in V4 neurons depending on the type of behavioral task[24]. The peak-normalized population spike rates in one monkey (animal S) show greater sustained activity than the other (animals P, Fig. 3b, c). This difference arose from differences in average absolute peak firing rate (monkey P mean 28.7 SEM 1.1 spikes/s, monkey S mean 24.6 SEM 0.9 spikes/s, $p < 0.01$, rank sum test) and average absolute sustained rates (monkey P mean 16.2 SEM 0.7 spikes/s, monkey S mean 18.5 SEM 0.6 spikes/s, $p < 0.01$, rank sum test). Principal component (PC) analysis of spike peri-stimulus time histograms of all neurons revealed that the monkey P has higher PC scores associated with third PC, which captured a transient peak response (Supplementary Fig. 7). To quantify neuronal modulation by attentional intensity, we computed a modulation index (neuronal $d'$) as the difference of $z$-scored firing rates (60–260 ms from sample onset) between high and low attentional intensities. The mean firing rate was significantly greater during the high intensity condition (neuronal $d'$ for all units, mean = 0.13, $p = 10^{-153}$, $n = 970$, $t$-test; Fig. 3d). Both single neurons and multiunit clusters showed significant modulation of firing rates by attentional intensity (neuronal $d'$ for single units, mean = 0.09, $p = 10^{-31}$, $n = 298$, $t$-test; for multiunit clusters,

mean = 0.14, $p = 10^{-127}$, $n = 672$, $t$-test). The attentional intensity effect was also significant for individual monkeys (monkey P, mean = 0.19, $p = 10^{-94}$; monkey S, mean = 0.09, $p = 10^{-82}$; Supplementary Fig. 8). In addition, of 970 units recorded in V4 from the two monkeys, significant intensity modulations ($p < 0.05$) were observed in 595 (61%) units. Thus, an isolated change in attentional intensity, affects responses in V4 in the absence of behavioral selectivity or response criterion changes.

It is expected that with increased reward size, animals' level of general arousal might also increase. Next, we wanted to test if the observed effects due to changes in attentional intensity can be fully explained by general arousal. It is believed that animal's general arousal and motivation decreases with time during a long experimental session which can be reflected by pupil size and the rate of aborted trials[16]. We separately analyzed behavioral $d'$, rate of aborted trials (fixation breaks), pupil area, and V4 spike counts for the early (first half) and late (last half) trials for small- and large-reward blocks within each session (Supplementary Fig. 9 and Supplementary Table 4). We then compared the modulations of these measures between attentional intensity and session timing. Attentional intensity significantly affected behavioral $d'$ ($p = 10^{-3}$) and the rate of aborted trials ($p = 10^{-3}$), whereas trial timing (early versus late) had no significant effects on behavioral $d'$ ($p > 0.05$) or aborted trials ($p > 0.05$). Thus, the level of attentiveness relevant to the task in a given type of reward block (small versus large) did not change detectably over longer intervals within a session irrespective of changes in satiation (motivation) and fatigue (or general arousal). In contrast, pupil area and V4 spike counts were significantly affected by both the attentional intensity ($p = 10^{-3}$) and trial timing ($p = 10^{-4}$). However, there were no significant effects of intensity-by-trial timing interaction on pupil area or spike responses ($p > 0.05$). Together, these results suggest that the attentional intensity

and general arousal can independently modulate common downstream targets such as autonomic sympathetic nervous system (pupil dilation) and cortical neuronal activity.

**Higher attentional intensity reduces neuronal response variability.** In addition to enhancing firing rates, increased attentional intensity reduced the trial-to-trial variability of spike rates of individual neurons (Fig. 3e). Following sample stimuli onset, the mean-matched Fano factor (the ratio of the variance of the firing rate to the mean; "Methods") dropped and remained significantly lower for high attentional intensity compared to low intensity (mean ± SEM Fano factor over 60–260 ms, for all units $0.881 \pm 0.001$ (small), $0.865 \pm 0.001$ (large), $p = 10^{-16}$, for single units $0.889 \pm 0.003$ (small), $0.867 \pm 0.002$ (large), $p = 10^{-7}$, for multiunits $0.875 \pm 0.002$ (small), $0.862 \pm 0.002$ (large), $p = 10^{-5}$, t-test; Fig. 3e). Attentional intensity did not have any effect on the Fano factor before the sample stimuli onset (mean ± SEM over −200 to 0 ms, for all units $0.970 \pm 0.003$ (small), $0.966 \pm 0.002$ (large), $p = 0.21$, for single units $0.986 \pm 0.003$ (small), $0.982 \pm 0.003$ (large), $p = 0.38$, for multiunits $0.961 \pm 0.005$ (small), $0.956 \pm 0.004$ (large), $p = 0.38$, t-test; Fig. 3e).

We also computed spike-count correlations over 200 ms during sample stimuli (60–260 ms) between pairs of simultaneously recorded neurons (Fig. 3f). Neuron pairs were binned based on their evoked responses (geometric mean of baseline subtracted spike rate, "Methods"). Intense nonselective attention reduced spike-count correlations (mean ± SEM, for all units $0.0616 \pm 0.0007$ (small), $0.0539 \pm 0.0006$ (large), $p = 10^{-60}$, for single units $0.0725 \pm 0.0026$ (small), $0.0616 \pm 0.0025$ (large), $p = 10^{-13}$, for multiunits $0.0605 \pm 0.0009$ (small), $0.0542 \pm 0.0009$ (large), $p = 10^{-20}$, t-test). Collectively, these results indicate that the attentional intensity modulation of V4 neurons is highly similar to those described for spatially selective attention[6,7,25].

**Spatial distribution of neuronal modulation.** The definition of attentional intensity used here depends on spatially uniform performance as measured by behavioral $d'$ across the hemifields. The task design does not rule out all forms of spatial selection, as the animals might have distributed their attention to other locations beyond the two stimulus locations tested. If so, V4 neurons could have been modulated by the spatially selective shifting of attention from those other locations to the two stimulus locations. Responses of V4 neurons with RF away from the focus of spatially selective attention are generally not modulated. Thus, an absence of spike modulation for the neurons with RF slightly off from the focus of attention would indicate a significant contribution of spatially selective attention in the observed attentional intensity-related neuronal effects. We examined the spatial distribution of attention by measuring the correlation between firing rate modulation and the proximity of V4 neurons' RFs and attended Gabor stimulus (Fig. 4). The proximity was estimated by the proportion of overlap between two-dimensional probability densities of neurons' RF and Gabor stimulus (Fig. 4a, b). We saw a wide distribution of RF-Gabor overlaps across recorded V4 units (median 22%, quartiles 12 and 30%; Fig. 4c). A monotonic increase (positive correlation) in spike rate modulation with the increase in RF-Gabor overlap would indicate spatially selective attention. Instead, we found no detectable change in neuronal modulation based on the RF-Gabor overlap (for all units, $\rho = 0.0002$, $p = 0.49$; single units, $\rho = -0.12$, $p = 0.97$; multiunits, $\rho = 0.04$, $p = 0.18$; Spearman correlation; bottom, Fig. 4d). There was no significant difference in spike modulation among groups of units with four equal intervals of RF-Gabor overlap ($F_{(3, 730)} = 0.67$, $p = 0.57$, ANOVA; top, Fig. 4d). This suggests that the firing rate increase with increasing attentional

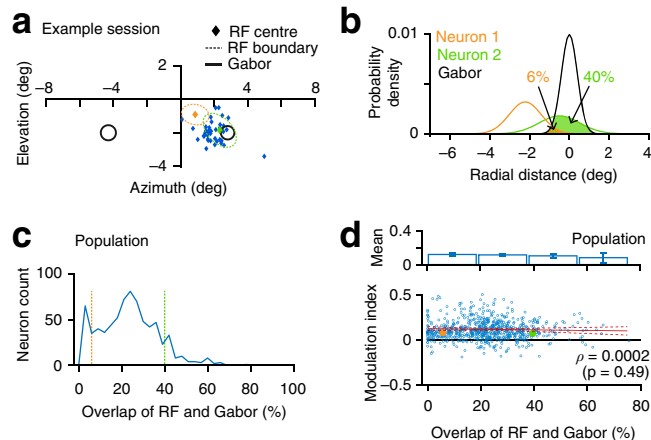

**Fig. 4 Spatial distribution of neuronal modulation associated with spatially nonselective attention.** **a** Centers of visual receptive fields (RF) of all recorded units in one example session. Black circles, Gabor sample stimuli (1 standard deviation (SD)). Dashed circles, size of RFs (1 SD) of two example neurons. **b** Overlaps of the probability densities of RFs and attended Gabor sample stimuli for two example neurons in (**a**) (see "Methods"). The proportion of overlap measures the proximity of the RF relative to sample stimuli. **c** Population distribution of the overlaps of RFs and Gabor stimuli across all units ($n = 731$). Dashed lines, two example neurons in (**a**, **b**). **d** Bottom, correlation between neuronal modulation (neuronal $d'$, Fig. 3d) and the proximity of RF and Gabor stimulus ($\rho = 0.0002$, $p = 0.49$; Spearman correlation, two-sided t-test). Red line, linear regressor (with 95% confidence intervals). Filled circles, example neurons in (**a**). Top, mean modulation indices of units grouped in four equal intervals of RF-Gabor overlaps. There was no difference among these groups ($F_{(3, 730)} = 0.67$, $p = 0.57$; one-way ANOVA). $n = 731$ units from monkey P and monkey S. Data are presented as mean with 95% confidence intervals.

intensity is distinct from previously described selective attention modulation of V4 spike rates.

Because we recorded from the same fixed multielectrode arrays over many sessions, it is possible that some units were sampled in more than one session. We investigated the effect of potential resampling by analyzing a subsample that included only one unit from each electrode across all recording sessions ($n = 159$ from two monkeys). For this conservative set, high attentional intensity increased spike rate (neuronal $d'$, mean $= 0.13$, $p = 10^{-25}$, $n = 129$, t-test; Supplementary Fig. 10) and reduced mean-matched Fano factor (mean ± SEM Fano factor, $0.914 \pm 0.004$ (small), $0.873 \pm 0.004$ (large), $p = 10^{-10}$) to extents that were indistinguishable from the whole population. Thus, the results are robust and independent of any multiple sampling that might have occurred.

**Trial-by-trial behavioral, physiological, and neuronal dynamics in response to reward modulation.** Many experimental covariates might closely follow the timing of changes in reward size used to manipulate attentional intensity. We therefore examined the dynamics of behavioral, physiological, and neuronal responses associated with reward changes. All the blocks of trials were first aligned to the first correct trial after block transitions. Trial-by-trial values of sample stimuli-evoked pupil area (physiological response) and spike counts were then averaged across blocks (Fig. 5 and Supplementary Fig. 13). In order to estimate the trial dynamics of behavioral performance, $d'$ at a given trial order (1–120) was computed using all behavioral responses at that trial order across blocks. The first reward in a block unambiguously signaled a block transition, which was otherwise unannounced.

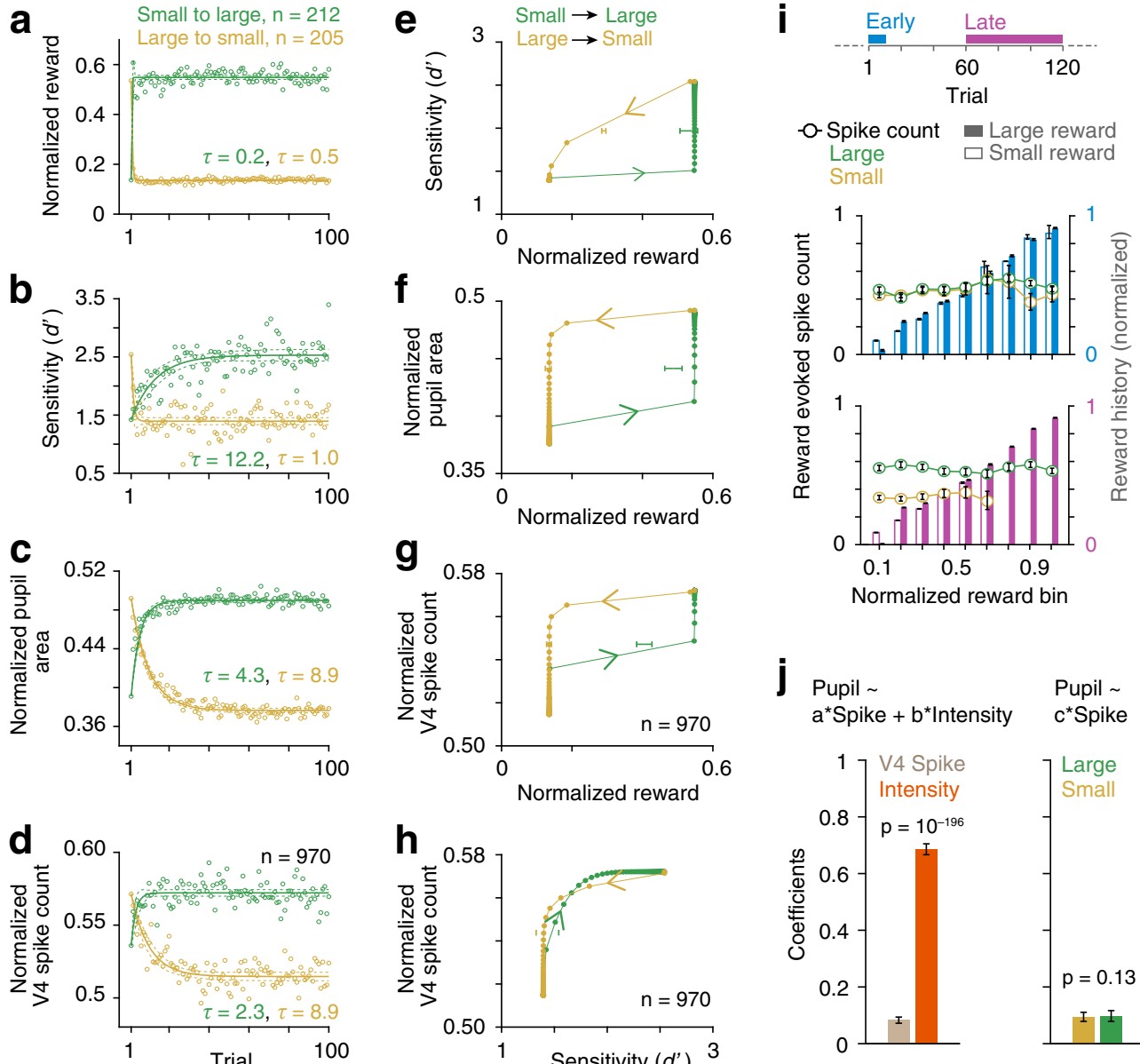

**Fig. 5 Directions of reward change have differential delayed effects on behavior and physiology.** Block-averaged single received rewards (**a**), behavioral sensitivity ($d'$) (**b**), normalized mean pupil area during sample stimulus period (**c**), and normalized V4 spike counts across all neurons ($n = 970$; (**d**)) in large ($n = 212$) and small ($n = 205$) reward blocks. Circles, observed data. Lines, single exponential fits. $\tau$, decay or rise constants. Trials are aligned with respect to the first correct trial following block transition. Dashed lines, 95% confidence intervals. **e**–**g** Hysteresis with transition of attentional intensity. Behavioral $d'$ (**e**), pupil area (**f**), and V4 spike counts (**g**) in response to reward changes from large to small and reverse. Horizontal error bars, mean ± SEM of reward size across blocks at the center value of y-axis variable. **h** V4 spike counts as a function of $d'$ changes. Vertical lines, mean $d'$ across blocks at the half-value of spike counts. **i** Top, analysis windows for reward-matched V4 spike counts in large- and small-reward blocks within first 10 trials at the beginning (middle, cyan) and last 60 trials at the end of blocks (bottom, purple). Bars, mean received rewards across trials binned between 0 and 1 (bin size = 0.2, overlap = 0.1; within session normalized) for two reward block conditions. Data are presented as mean with 95% confidence intervals (bootstrap, $n = 10^4$). Circles, mean normalized spike counts ($n = 970$) on immediate next trial. **j** Left, linear regression between mean pupil area (0–400 ms), spike counts (60–260 ms), and reward conditions (high and low attentional intensities) across all trials were fit separately for each neuron. Population mean ± SEM of fitted coefficients of intensity was significantly higher compared to the coefficients of spike counts ($p = 10^{-196}$; rank sum test) for neurons with significant fit ($p < 0.05$, $n = 625$; F-test). Right, two separate linear regression fits between pupil area and spike counts for large- and small-reward trials. Mean ± SEM of fitted coefficients of spike counts for the two reward conditions for neurons with significant fit ($p < 0.05$, $n = 334$; F-test) were not different ($p = 1.3$; rank sum test).

Block-averaged single-trial reward values received by the animals approximated step functions (single exponential fit: $\tau = 0.2$ and 0.5 trials, respectively, for transitions from small to large ($n = 212$ blocks) and from large to small ($n = 205$ blocks), Fig. 5a). Behavioral $d'$ closely tracked reward changes for the transition from large to small ($\tau = 1$). However, the $d'$ was much slower ($\tau = 12.2$) for transitions from small to large (Fig. 5b). As expected, trial constants of percent correct, followed a time course similar to $d'$ (large to small, $\tau = 1$; small to large, $\tau = 9.0$; Supplementary Figs. 11 and 12). In contrast to task performance,

pupil area and population mean V4 spike responses (60–260 ms from sample onset) had the same slow decays for large-to-small transitions ($\tau = 9$ for both) and similar, faster rises for small-to-large transitions ($\tau = 4$ and 2, respectively, for pupil area and spikes, Fig. 5c, d). Similar to previous studies[16], we also found that the changes in reward size differentially affected trial-by-trial rate of aborted trials depending on the direction of reward change (Supplementary Fig. 14). Immediately after the block transition, the aborted trials slowly increased ($\tau = 4.2$) for transitions from large to small compared to transitions from small to large ($\tau = 0.1$). However, as the trials progressed abort trial rate gradually decreased in small-reward blocks. Taking into account the dissimilar early aborted trial rates between small- and large-reward blocks did not alter qualitatively trial-by-trial dynamics of behavioral $d'$, pupil area, and V4 spike counts as a function of time rather than trial count (Supplementary Fig. 15). Time constants of these behavioral, physiological, and neurophysiological variables were closely proportional to the trial constants as measured from trial-by-trial dynamics as a function of number of trials.

The discrepancies between the effects of two directions of reward changes on neuronal firing, pupil area, and behavior can be clearly seen in hysteresis plots (Fig. 5e–g). These response transitions show memory effects where an equivalent change in the response requires significantly different amount of reward change depending on the change direction. Behavioral $d'$ tracks transitions from small-to-large reward size more slowly than transitions from large-to-small reward size (Fig. 5e). In contrast, pupil area and neuronal responses track transitions from small-to-large reward size more quickly than transitions from large to small. The normalized reward value at which the values reach their mid-point differ significantly for large-to-small and small-to-large transitions for all three measures (horizontal bars in Fig. 5e–g, mean ± SEM, $p = 10^{-31}$, $t$-test). Similar neuronal hysteresis is also seen in motion perception and contrast detection in humans where stimulus history affects perceptual detection threshold[26,27]. The different dynamics between $d'$ and spike rate were almost complementary (Fig. 5h), with spike rate leading for low-to-high transitions and lagging for high-to-low transitions of $d'$.

While the average spike rate in area V4 changed slowly following block transitions, the averages might obscure trial-to-trial changes in neuronal activity that tracked the reward received on the previous trial. Because rewards for hits and CRs typically differed (Supplementary Fig. 1), we could measure dependency of spike counts on the previous reward. We sorted trials into different bins depending on the normalized reward size on the preceding trial and measured spike counts of each neuron (see "Methods"). We further separated the trials based on whether they occurred immediately after block transitions (within first one to ten trials from the first correct response after the reward switch) or during steady state within a block (final 60 trials, 61–120) (top, Fig. 5i). Reward value on the previous trial did not affect spike counts ($F_{(8, 12801)} = 1.46$, $p = 0.164$ for repeated measured factor reward value (nine levels); $F_{(1, 12801)} = 3.72$, $p = 0.0537$ for between group factor, block-reward condition (large and small)) (middle, Fig. 5i). The lack of dependency of spike responses on immediate expected reward remained unchanged even during the late, steady state portion of blocks (late trials), when a large effect of block-reward size was seen ($F_{(8, 10752)} = 1.18$, $p = 0.307$ for reward value; $F_{(1, 10752)} = 7.78$, $p = 0.005$ for block-reward condition) (bottom, Fig. 5i). Thus, the strong V4 response modulations we saw were dominated by slow effects associated with reward history and not due to immediate expected reward.

The similar dynamics of pupil area (Fig. 5c) and spike rate (Fig. 5d) suggest that a larger pupil area associated with high attentional intensity might enhance effective retinal illumination,

which might elevate V4 neuronal spiking. To test this possibility, we quantified how well pupil area correlates with V4 spike rates and attentional intensity ("high" or "low") for all neurons using linear regression (see "Methods"). Regression coefficients for attentional intensity were more than eightfold higher than the coefficients for spike rate (mean ± SEM, spike rate, 0.08 ± 0.005; attentional intensity, 0.68 ± 0.01; Wilcoxon rank sum test (WRS), $n = 625$, $p = 10^{-196}$; Fig. 5j, left). The weak relationship between pupil area and spike rate was similar for the two reward conditions, when mean pupil area was fit separately for small- and large-reward trials (mean ± SEM, small reward, 0.094 ± 0.008; large reward, 0.097 ± 0.01; WRS test, $n = 334$, $p = 0.13$; Fig. 5j, right). We additionally measured the temporal relationship between single-trial V4 spike trains and pupil area dynamics using cross-correlation and spike-triggered averaged (STA) pupil area for each neuron (see "Methods"). None of the neurons show any significant correlation between peri-stimulus time histograms (PSTHs) of pupil area and spike rates (binned at 10 ms) over the analysis window from −350 to +350 ms from sample onset (Supplementary Fig. 16). Similarly, STA-pupil area that measured the extent to which individual spikes were aligned with pupil area did not show any significant relationship between single spikes of individual neurons and pupil area (absent of STA-pupil area > 0 at negative lags; Supplementary Fig. 17) compared to subcortical brain structures including locus coeruleus and superior colliculus[28]. Together, these results suggest that V4 spike activity and the autonomic sympathetic system that regulates pupil dilation are both strongly modulated by common top-down cognitive states rather than a direct influence of changing retinal illumination on V4 spike activity within the limit of observed pupil area change.

**Single-trial encoding of attentional states in area V4.** To understand better how V4 neurons encode a cognitive state of enhanced attentional intensity, we fitted single-trial spike counts of every neuron using a GLM as a function of experimental predictor variables (see "Methods"). In a detailed GLM (complete model; Fig. 6a), we used 13 predictor variables: orientation of sample stimuli inside RF; mean pupil area during the sample period; the saccadic choice; and a reward history that consisted of reward outcomes in each of the previous ten trials. We also compared the performance of this complete model with two reduced GLMs (Supplementary Fig. 18) that included either reward history (reward model) or the pupil area (pupil model) alone (Fig. 6a). Fitted single-trial spike counts in small- and large-reward conditions from cross-validation test data set for an example neuron are shown in Fig. 6c. Model performance was measured by pseudo $R^2$ values (see "Methods"). Most neurons (78%, 757/970) were significantly fit ($p < 0.05$, $F$-test) with the complete model. Fewer neurons had significant fits for the reduced models (reward model, 12%, 114/970; pupil model, 50%, 483/970; Fig. 6b). Figure 6d shows population-averaged spike counts of all fitted neurons estimated from the complete GLM with observed responses overlaid for the cross-validated test data set. We next compared the relative contributions of the experimental variables on spike responses for each neuron as measured by predictor importance (PI), which is the normalized magnitude of fitted coefficients (see "Methods"). Stimulus orientation has the strongest contribution to the spike counts, which is expected because most V4 neurons are orientation selective (PIs of individual neurons, Fig. 6e; mean PI across neurons, Fig. 6f). More interestingly, single-trial pupil area also strongly predicts spike response compared to reward history and saccade choice. Instantaneous pupil area remained a strong predictor when we compared with constant levels of attentional intensity ("high" and

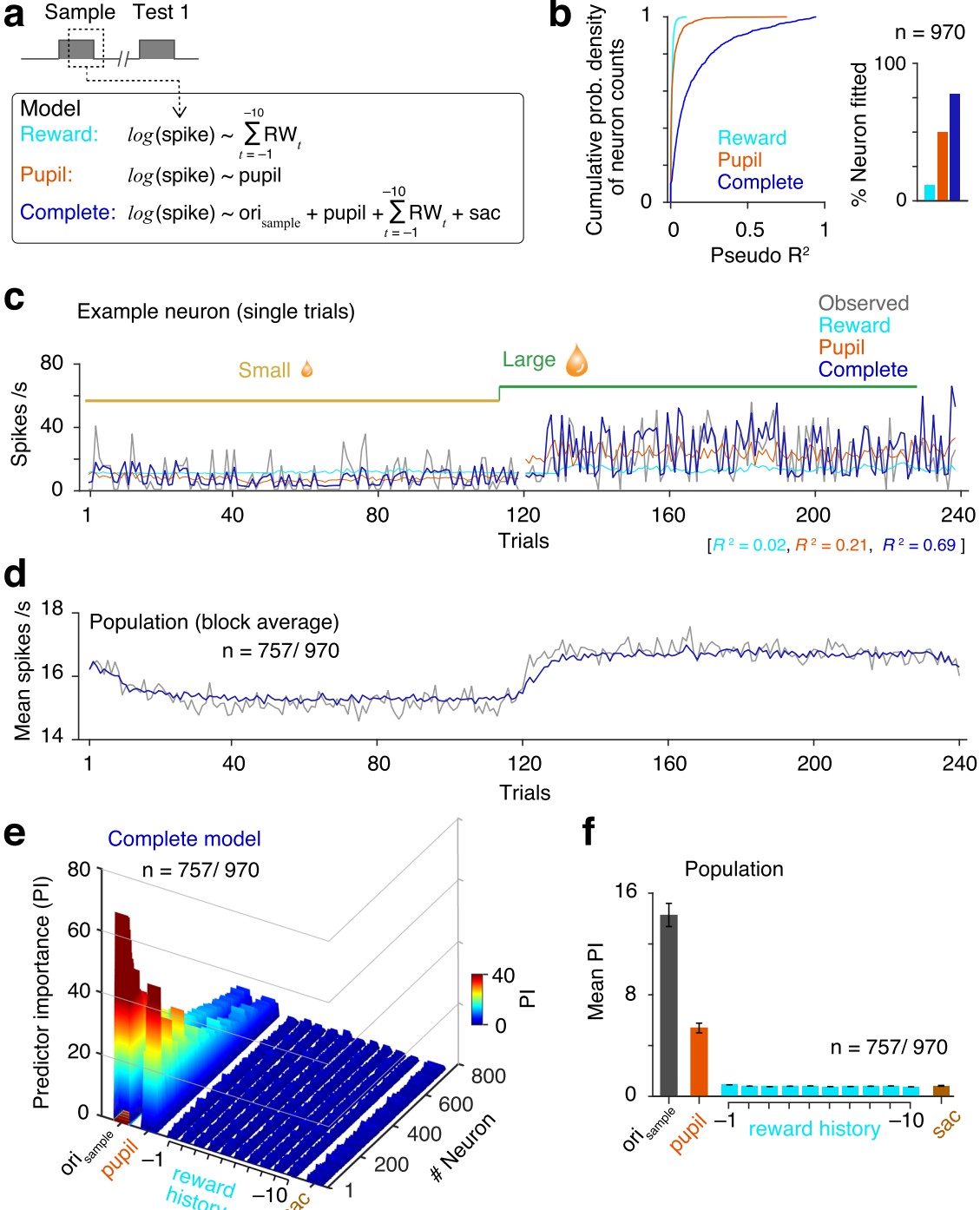

**Fig. 6 Encoding of cognitive states by single-trial spike counts. a** Top, generalized linear models (GLM) on single-trial spike counts during sample stimuli period (60–260 ms from sample onset). Spike counts were fit with three different GLMs. Two were reduced models, based on either reward history (ten immediate past received rewards; reward model) or mean pupil area over 400 ms following stimulus onset (pupil model). A more detailed model contained multiple experimental and cognitive variables that included reward history (past ten trials), pupil area, saccade response, and Gabor stimulus orientation (complete model). **b** Left, cumulative probability densities of pseudo $R^2$ values (see "Methods") of single neurons measure goodness of fit for different GLMs. Right, fraction of the neuron population that were fit significantly better ($p < 0.05$, $F$-test) compared to a null model (a constant) (reward model, 114/970 (12%); pupil model, 483/970 (50%); complete, 757/970 (78%)). **c** Observed and model-predicted spike counts on single trials from cross-validation test data set for an example neuron in two representative reward blocks. **d** Population-averaged observed and fitted (complete model) spike counts. Gray, observed; blue, complete model. **e** Comparing predictor importance (PI) that measures contributions of different predictor variables estimated by absolute standardized predictor coefficient values. Colormap represents PIs of all neurons that were fit with the complete model ($n = 757$, $p < 0.05$; $F$-test). Neurons were sorted based on the pseudo $R^2$ values. **f** Population-averaged PIs well fit neurons ($n = 757$) presented as mean with 95% confidence intervals (bootstrap, $n = 10^4$).

"low") as well as block-averaged single-trial pupil area in two alternate models (PI, mean ± SEM, constant attentional intensity, 2.3 ± 0.07; blocked-averaged pupil area, 2.7 ± 0.09, instantaneous pupil area, 5.37 ± 0.19; ANOVA, $F_{(2, 2195)} = 166.31$, $p = 10^{-67}$; Supplementary Fig. 19).

Together, these suggest that even though reward history is the primary external motivator for the monkeys, V4 spike responses to the sample stimuli encode the level of instantaneous attentional intensity more strongly than recent reward outcome value or saccade choices.

It is possible that reward history has more weight around the time of block transitions, when expectations change rapidly[16]. Because most of the trials within blocks in our task occur long after a transition (steady state), this weight would be diluted. To address this, we also fit spike counts from only the first 20 trials after block transitions with the same complete GLM as shown in Fig. 6a (Supplementary Fig. 20). As with the complete data set, stimulus orientation and pupil area contribute much more strongly to V4 spike counts than does reward history. Although both the pupil area and spike counts are not directly related, they are linked to the top-down attentional intensity in a similar way. Thus, the state of spatially nonselective attention as represented by pupil area is encoded in single-trial spike counts during the sample stimuli period.

**Dynamic representation of the detection of behaviorally relevant sensory signal in area V4.** Although reward representation in V4 is weak during the sample stimuli interval, stronger signals might emerge closer to the behavioral response. We therefore separately fit spike counts during the choice interval (test 1 interval) using a similar complete GLM. We used only those trials for which test 1 stimulus appeared inside RFs and the animal did not initiate a saccade until after the analysis period (which was 60–260 ms from test 1 stimulus onset; 44.4% (22,238/50,038) of the trials) (Supplementary Fig. 23). The stimulus feature variable in this model was the product of orientation tuning filter and test 1 stimulus orientation, as there were four different orientations of test 1 stimulus. The remaining variables were mean pupil area during the test 1 presentation, reward outcomes in each of the previous ten trials and the saccadic choice.

Reward history remained a poor predictor of single-trial spike counts compared to pupil area (Supplementary Fig. 23). Notably, the saccade choice variable is also a strong predictor of spike responses for the same neuronal population. Previous reports have shown that spike responses of V4 neurons enhance before initiating a saccade toward the RF location[17,18]. Among all the saccade sensitive neurons ($n = 256$, significant saccade coefficient, $p < 0.05$) about two-thirds were more active before saccades (saccade-preferred neurons; saccade coefficient > 0; $n = 149/256$) and the rest were less active (saccade anti-preferred neurons, saccade coefficient < 0; $n = 107/256$). Furthermore, based on behavioral outcomes (hit, miss, CR, and FA), we sorted observed spike counts on single trials during sample and test 1 stimulus periods for these two neuron types, saccade preferred and saccade anti-preferred (Supplementary Fig. 23). A true encoding of saccade choice by these neurons predicts indistinguishable spike response between hits and FAs (and between CRs and misses). Instead, spike responses of saccade-preferred neurons increase for hits and misses (nonmatch trials) compared to CRs and FAs (match trials). Similarly, responses of saccade anti-preferred neurons increase for CRs and FAs (match trials) compared to hits and misses (nonmatch trials). This suggests that saccade choice is not a suitable predictor variable in accounting for single-trial spike counts during the test 1 interval; rather "stimulus orientation change" can be a better predictor parameter.

We next fit the same data set of spike counts during test 1 stimulus with a complete GLM as mentioned above except, the saccadic choice variable was replaced with stimulus orientation change (Δori) (top, Fig. 7a). The variable Δori represents the trial type: match or nonmatch trial. Covariance between single-trial absolute test 1 stimulus orientation and Δori is small (mean partial correlation 0.005) across sessions due to small Δori values (16°–32°). Furthermore, we used only the neurons with very weak correlation coefficients (<0.2) between model-predicted spike counts due to stimulus feature (product of orientation tuning filter and test 1 orientation) and Δori to rule out stimulus orientation-related effects on estimates of Δori coefficient. V4 spike counts robustly encode information of single-trial Δori (Fig. 7a, b). Based on significant Δori coefficients ($p < 0.05$), two populations of neurons were classified, Δori preferred (Δori coefficient > 0, $n = 130/521$) and Δori anti-preferred (Δori coefficient < 0, $n = 109/521$). As expected, orientation change differentially modulates mean spike counts during the test 1 interval in these two neuron subpopulations in accordance with the change in stimulus orientation (two factors repeated measured ANOVA, $F_{(1, 237)} = 15.33$, $p = 10^{-4}$ for neuron types (Δori preferred, Δori anti-preferred); $F_{(3, 711)} = 7.83$, $p = 10^{-4}$ for repeated measured factor response choices (hit, miss, CR, and FA)) (right, Fig. 7c).

Finally, we tested the detection accuracy of stimulus orientation change from observed V4 spiking activity during the test 1 stimulus by using the fitted model on a single trial in a tenfold cross-validation test data set (see "Methods"). A random pair of trials, one from nonmatch (hit or miss) and another from match (CR or FA) without replacement, were selected. For each of these two trials, we evaluated the likelihoods of observed spike counts to be consistent with spike counts under the two possible orientation changes for each fitted neuron within a session. Sum of the log-likelihood ratios across simultaneously recorded neurons amounts to the predicted probability of a realization of orientation change on that trial. The discrimination of an orientation change from no change was considered correct only when the decoded orientation changes in both trials of the selected pair matched the observed data. Figure 7d shows the model-fitted population-averaged spike counts and decoded task variables, orientation change (top, using Δori GLM) and saccadic choice (bottom, using saccade GLM as described in Supplementary Fig. 23) on 20 pairs of randomly selected trials in an example session. Similar to saccade prediction, we also tested accuracy in discriminating response choices, i.e., hit from miss or FA from CR from observed spike counts during test 1 using the same saccade GLM. Prediction accuracy of orientation change (using Δori GLM) is better than both saccade and choice (hit versus miss or CR versus FA) predictions (using saccade GLM) across all sessions (mean ± SEM, Δori, 68.5 ± 2.8%, saccade, 50.0 ± 1.4%, choice, 34.2 ± 0.8%; $p = 10^{-25}$, $F_{(2, 69)} = 154.2$, ANOVA; Fig. 7e). Decoding performance of Δori is better compared to saccade irrespective of the number of neurons used for decoding within a session (Fig. 7f). Together, these encoding and decoding models offer powerful methods in isolating task-relevant variants and reading out perceptual or decision signals from a population of neuronal spikes.

## Discussion

We used simultaneous multineuron recordings and single-trial-generalized linear modeling to uncover how visual area V4 activity represents information related to intensity of attention, perceptual detection, reward expectation, and the saccadic motor action relative to sensory stimulus. First, we found that isolated changes in spatially nonselective attention between two locations modulate the activity of neurons in V4. Previous reports on

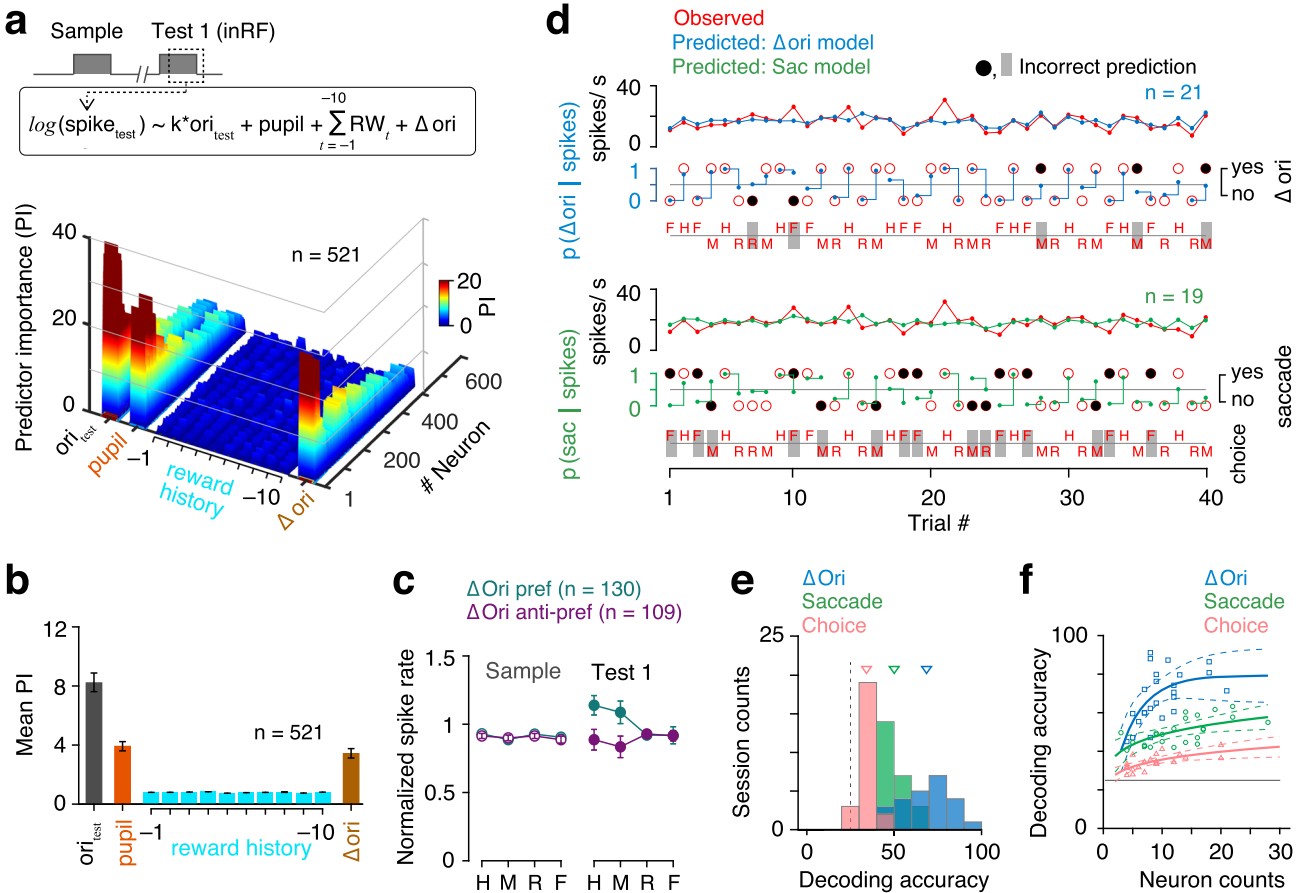

**Fig. 7 V4 spikes encode change in stimulus orientation immediately before response selection. a** Top, Δori model: GLM of spike counts during test 1 interval (60–260 ms from test 1 on). Predictor variables: product of test 1 stimulus orientation and neuron's orientation tuning filter, pupil area, reward history, and orientation change (match or nonmatch trial). Only the trials when test 1 stimulus appeared inside RF of recorded neurons and monkey did not initiate any saccade before analysis period (44.4% of the total trials). Bottom, contributions of predictor variables to spike counts of single neurons. **b** Population-averaged predictor importance presented as mean with 95% confidence intervals (n = 521). **c** Observed normalized mean (with 95% confidence intervals) V4 spike rates of Δori responsive neurons during sample and test 1 stimulus presentations for different behavior choice trials. H hit, M miss, R correct rejection, F false alarm. Neurons with significant Δori coefficient (p < 0.05; two-sided t-test) are classified as Δori preferred (coefficient < 0, n = 130; two-sided t-test) or Δori anti-preferred (coefficient < 0, n = 109; two-sided t-test) based on the GLM fits. **d** Decoded task variables on randomly selected pair of trials from cross-validation test data set in an example session. GLM fitted coefficients and observed single-trial V4 spike counts are used to estimate probability of Δori (Δori model, (**a**)) and probability of saccade (saccade model, Supplementary Fig. 12). Open red circle, observed response. Filled black circles, incorrect prediction. Dot, predicted probability of a response. **e** Decoding accuracy of response choice, saccade and Δori for cross-validation test data sets across all sessions (n = 24) using GLM fits. Triangles, population means. Dashed line, 25% chance level. **f** Decoding accuracy as a function of number of neurons used for decoding across sessions (n = 24). Markers, individual session. Solid lines, second-order polynomial fits. Dashed lines, 95% confidence intervals.

modulation of V4[14,15] activity with "effort" by changing task difficulty is thought to represent the intensive aspect of attention[13]. The changes in effort in these studies[14,15] were always conflated with changes in attentional selectivity, which is known to strongly modulate V4 neuronal activity[5–7,25]. The current data show conclusively that V4 neurons are modulated by attentional intensity in a manner similar to spatially selective attention-mediated effects, including an increase in spike rate and reductions in both individual variability as well as pairwise spike-count correlations. Unlike selective attention, the spike rate modulation with attentional intensity is spatially nonselective and independent of the neuron's RF location. Previous studies have reported that visual spatial attention can be split into multiple foci to suppress task irrelevant distractor information[29,30]. Absence of any distractor in the current task and spatially nonselective neuronal modulation with changes in attention intensity eliminates the possibility of split spatial attention-mediated effects in the results. Second, changes in reward size result in correlated

changes in the averaged cortical responses, pupil area—a physiological signature of attention and arousal—, and behavioral performance (d′). However, neuronal responses, pupil area, and behavioral d′ dissociate markedly at block transitions, exhibiting distinctly different degrees of dependency on trial history and single-trial dynamics. Third, sensory stimulus and pupil area strongly predict V4 spike counts throughout individual trials, as revealed by single-trial information encoding. However, V4 spike counts are little affected by recent reward history. The detection of sensory stimulus change emerges to be strongly represented within the spike counts immediately before the behavioral choice when animals make the decision. Together, these results isolate the dynamics of multiplexed representations of cognitive and experimental signals in V4 which can be crucial for performance in attention-demanding tasks.

In these experiments, we controlled attentional intensity using changes in reward size. Some formulations in psychology and neuroscience term the intensive component of attention as

"effort" which is uniquely associated with changes in task difficulty, and view it to be a specific type of "arousal" but distinguishable from other forms of arousal such as that induced by stress, novel bottom-up stimuli, and drugs[13] (but see[31]). It is possible that the brain might incorporate distinct forms of arousal. For example, while increased arousal is typically viewed as enhancing task performance, elevated generalized arousal caused by stressful bottom-up stimuli is typically associated with poor performance, as measured by lower response criterion with faster and less accurate responses[32]. It is possible that the phenomena of nonselective attentional intensity we observed with reward manipulation either represent mechanisms that work in parallel with overall arousal or specific components of overall arousal similar to "effort" induced by task difficulty. Moreover, neurophysiological evidence suggests that common neural pathways are activated by either changes in expectation of reward or changes in task difficulty[33]. Understanding of the relationship(s) between attentional intensity, effort, and arousal might be refined through future experiments with precise and independent control of these cognitive factors on simultaneous tasks. Other important questions to be addressed concern the circuit, cellular, and molecular mechanisms that mediate attentional intensity. These could involve activation of diverse neuromodulatory systems such as norepinephrine, acetylcholine, serotonin[34–39], and inputs from other brain regions, including direct anatomical projections including the amygdala[40,41].

In the current experiments, pupil area provides a reliable physiological correlate of the dynamics of attentional intensity[42]. Various other cognitive processes have been linked with pupil size changes, including memory, decision-making, and emotions[11,43–46]. Changes in pupil size are related to the activity of neurons in the locus coeruleus (LC)[28], which projects extensively to attention processing areas including visual cortex. A recent study has shown that V4 spiking activity strongly covaried fluctuations in pupil size association with spontaneous slow drift in task performance[47]. Like V4, many other brain areas show correlated spike activity with pupil area, including structures implicated in attention[28]. A noradrenergic neuromodulation of V4 neurons might mediate such brain-wide attentional state representation. It might be valuable to learn whether changes in pupil size and behavioral performance associated with attentional intensity depend on changes in LC activity or are instead mediated in part or wholly by other structures.

When reward sizes increase, the rise of $d'$ is slow relative to the increase in V4 spike rate and pupil area. In contrast to reward increases, performance drops quickly with decreased rewards even though pupil area and neuronal response remain elevated (i.e., delayed decay, Fig. 5). Dissimilar effects of reward change directions on behavior are often seen across species[48,49]. Independence of V4 neuronal activity from single-trial behavioral performance is consistent with the notion that transitions between cognitive states occur with different dynamics in downstream structures that read out perceptual detection and weigh anticipated outcomes against the cost of initiating and executing action[16,50]. It remains to be determined how much subjects can control the dynamics of their behavioral responses to changes in reward expectation. The observed hysteresis in V4 spike responses with reward changes represents a type of memory. Hysteresis at the input–output level of a single neuron can improve memory capacity and retrieval properties in the presence of noise in a synchronous network of neurons with nonlinear threshold[51]. Indirect evidence of hysteresis in single cortical neurons comes from the fact that spike train history is an important predictor of individual spikes[52,53]. Spike counts on reward-matched trials are indistinguishable between the two reward conditions immediately after the reward switch and become different only in later trials when the animal's attentional

intensity attains a steady level (Fig. 5i). These results together with reward-spike hysteresis (Fig. 5g) strongly support a dependency of V4 neuronal activity on history of attentional intensity. Previous trial history of task-relevant signals has also been reported to strongly modulate PFC neurons[54]. Neuronal hysteresis during visual perception has been suggested to arise from cooperativity between interconnected neurons[27,55]. Future experiments are required to dissociate the role of single cell excitability and connectivity among neurons on neuronal hysteresis in V4. Absence of reward modulation (immediate past trial outcome) of spike counts on reward-matched trials in trials immediately after block transitions is inconsistent with immediate reward representation as reported in previous studies[16,56].

The relationships between reward expectation, behavioral $d'$, and V4 spiking were examined in a previous study[16] that found the expectation of a large reward increases spike rates of V4 neurons relative to expectation of a low reward, even when $d'$ is unchanged across those conditions. Consistent with this, we found that the $d'$ was slower to respond to changes in reward schedule than was V4 spiking. The early study randomly interleaved different rewards conditions on a trial-by-trial basis, whereas our task contingencies were entirely predictable over blocks of 120 trials. We would expect to see results qualitatively similar to the previous report[16] if we randomly interleaved trials. Unlike previously observed positive correlation between absolute reward size expectation and V4 spiking[16], we found weak representation of reward history in single-trial V4 spike counts using both reward-matched spike-count analysis and GLM-based encoding models. The reward representation remained weak even immediately after block transitions when the knowledge of reward expectancy was crucial in motivating the animals to alter their level of exerted attentional intensity and hence the behavioral performance. In our study, the reward on each trial varied around a mean within each reward schedule (Supplementary Fig. 1), whereas fixed rewards were used in the earlier study. Our additional reward-variance provided negligible covariances between reward size and other task variables, and was crucial for reliable isolation of multiplexed information on single trials using GLM. Prior application of GLM-based encoding in primate cortical areas has identified dependencies of correlated spikes on multiplexed sensory, cognitive, and motor signals[52,57]. Performance of our model fit can be further improved by taking into account of spike history or interaction with other units within the network[52].

Previous studies have described that V4 spiking increased when an animal intends to make a saccade toward the neuron's RF[17,18,58,59]. However, single-trial encoding shows that the observed correlation between V4 spike counts and saccade initiation is primarily due to perceptual signal detection and not saccade preference. The lack of a slow latency pre-saccadic signal in V4 spikes based on the spike counts over the 200 ms stimulus period does not fully rule out the possibility of fast pre-saccadic modulation that is shorter than 200 ms. Ablation of V4 has revealed no effects on saccade amplitude or latency to visual targets[60,61]. In partial agreement with some of these findings, our results confirm the role of V4 in single-trial perceptual detection of behaviorally relevant signals (orientation change) but reject a direct contribution to saccade choice. High covariance between signal detection and behavioral choice in previous studies may have limited the detailed isolation of information coding of perception from decision. In the present study, we identified two types of response populations in V4 that are differentially modulated with changes in behaviorally relevant signals. Future work at the microcircuit level is required for detailed characterization of these two cell populations with respect to their inputs and recurrent connections. Understanding the finer dynamics of

cognitive task variants within a single trial would require modeling at the time scales of single spikes[52]. A perceptual task with a continuous stream of visual stimuli would be ideal to evoke a sufficient number of spikes throughout the task period for studying visual areas like V4 that have low activity in the absence of visual stimulation.

Overall, our experimental framework provides new evidence revealing details of neuronal correlates of nonselective attentional intensity in visual cortex with reference to physiological and behavioral state of attentiveness. This approach can be extended to other brain areas to better understand how different cognitive signals are distributed and read out in mediating decision and action selection.

## Methods

**Subjects and surgery.** Two adult male rhesus monkeys (Macaca mulatta, 13 and 9 kg) were implanted with a titanium head post using aseptic surgical techniques before training began. After the completion of behavioral training (3–5 months), we implanted a 10 × 10 array microelectrodes with 400 μm spacing (Blackrock Microsystems) into dorsal visual area V4 of one hemisphere, between lunate and superior temporal sulci (Supplementary Fig. 5). The array was placed more dorsal with respect to the inferior occipital sulcus (ios) in monkey S compared to monkey P. Polar angles of the recorded units' RFs differed in the two monkeys (mean ± SEM, monkey P, 132.4° ± 2.6°; monkey S, 108.8° ± 3.2°; $p = 10^{-11}$, rank sum). Eccentricities of RFs in the two animals did not differ (mean ± SEM eccentricity, monkey P, 2.8° ± 0.1°; monkey S, 2.7° ± 0.1°; $p = 0.18$, rank sum). Similarly, RF sizes between animals were not different (mean ± SEM RF size (1 sigma), monkey P, 1.1° ± 0.05°; monkey S, 1.2° ± 0.06°; $p = 0.76$, rank sum) (Supplementary Table 3).

**Behavioral task.** During training and neurophysiological recording, the monkey was seated in a primate chair facing a calibrated CRT display (1024 × 768 pixels, 100 Hz refresh rate) at 57 cm viewing distance inside a darkened room. Binocular eye position and pupil area were recorded at 500 Hz using an infrared camera (Eyelink 1000, SR Research). Trials started once the animal fixated within 1.5° of a central white spot (0.1° square) presented on a mid-level gray background (Fig. 2a). The animal had to maintain fixation until its response at the end of the trial. After a fixation period of 400–800 ms, two achromatic Gabor sample stimuli appeared for 200 ms, one in each visual hemifield. After a variable delay of 200–300 ms, a Gabor test stimulus (test 1) appeared for 200 ms at one of the two target locations, randomly selected with equal probability. The test stimulus was identical to the preceding sample stimulus, except for its orientation. On half of the trials, the test 1 stimulus had a different orientation (nonmatch trial), and the monkey had to make a saccade to that target location to receive an apple juice reward. On the remaining half of the trials, the test 1 stimulus had the same orientation as the corresponding sample stimulus (match trial), and the monkey had to maintain fixation until a second test stimulus with a different orientation (test 2, 200 ms) appeared in the same location after an additional delay interval of 200–300 ms. The monkey then had to saccade to that target within 500 ms (monkey P) and 470 ms (monkey S) after the appearance of the test stimulus to get a reward. Intertrial intervals varied from 2–3 s. Stimuli were presented always in the lower hemifields at 2°–4° eccentricity. Gabors were odd symmetric with the same average luminance as the background. Spatial frequency, size, and base orientation of Gabor stimuli were optimized for one of the neurons recorded each day, and remained unchanged throughout each session (left, azimuth −2.5° to −5.3°, elevation 0.0° to −3.5°, sigma 0.35°–0.60°, spatial frequency 0.6–3.5 cycles/°; right, azimuth 2.8°–5.3°, elevation 0.0° to −2.5°, sigma, 0.40°–0.65°, spatial frequency 0.7–3.0 cycles/°). On every trial, the orientation of the sample stimuli randomly took one of two values, base orientation or orthogonal. Orientation change (difficulty) remained fixed within a session and varied across sessions between 24° and 32° for monkey S and 16° and 20° for monkey P.

Attentional intensity was controlled over blocks of trials by changing reward volumes for correct responses over a four- to five-fold range. Every 120 trials, reward size alternated between large and small values without any prior cue to the animal. Reward sizes for hits (correct response in nonmatch trial) and CRs (CRs in match trial) were adjusted as needed to encourage the animal to maintain a behavioral criterion close to 0[6]. Although, trial-averaged criterion within a reward condition remained 0, there was a transient change in criterion immediately after the block transition (reward switch) which then approached 0 over a few trials (Supplementary Fig. 12). Distributions of reward sizes are shown in Supplementary Fig. 1. Behavioral task was controlled using custom-written software (Lablib)[62].

**Electrophysiological recording and data collection.** Extracellular neuronal signals from the chronically implanted multielectrode array were amplified, bandpass filtered (250–7500 Hz), and sampled at 30 kHz using a Cerebus data acquisition system (Blackrock Microsystems). We simultaneously recorded from multiple single units as well as multiunits (563, monkey S; 407, monkey P) over 24 sessions

(15 for monkey S, 9 for monkey P). Before each experimental session, we mapped RFs and stimulus preferences of neurons, while the animal fixated. These RFs were used to optimize the stimulus parameters. Spikes from each electrode were sorted offline (Offline Sorter, Plexon Inc.) by manually well-defining cluster boundaries using principal component analysis (PCA) as well as waveform features. Well isolated clusters based on J3 statistics were classified as single units[63]. Single units were classified from multiunits based the isolation quality of unit clusters. The degree to which unit clusters were separated in 2D spaces of waveforms features (first three PC, peak, valley, energy) was measured by J3 statistics and Multivariate Analysis of Variance (MANOVA) F statistic using Plexon Offline Sorter (Plexon Inc.). A unit cluster of MANOVA $p$ value of <0.01 was considered as single unit which indicates that the unit cluster has a statistically different location in 2D space, and that the cluster is statistically well separated. J3 measures the ratio of the average distance between points within clusters to the average distance between clusters. It takes a maximum value for compact, well-separated clusters (for single units, mean J3 = 3.70, SEM = 0.16; for multiunits, mean J3 = 1.02, SEM = 0.02). We analyzed only those units that were stable throughout recording sessions, which lasted for 3–5 h. That duration limited the number of units per session. Not every electrode provided useful data during each session (monkey P, 37–53 units/ session; monkey S, 21–51 units/session).

## Data analysis

*Behavioral performance.* All completed trials (120 trial per block) were included in our analysis. Behavioral performance accuracy was expressed by behavioral sensitivity or $d'$ adapted from the "Signal detection theory" model[64]. Sensitivity measures subjects' ability to detect a signal from a noise independent of the response bias (or criterion). Behavioral sensitivity ($d'$) and criterion ($c$) were estimated from hit rates within nonmatch trials and FA rates within match trials as:

$$d' = \Phi^{-1}(\text{hit rate}) - \Phi^{-1}(\text{FA rate}) \qquad (1)$$

$$c = -\frac{1}{2}[\Phi^{-1}(\text{hit rate}) + \Phi^{-1}(\text{FA rate})] \qquad (2)$$

where $\Phi^{-1}$ is inverse normal cumulative distribution function. We measured average $d'$ and $c$ within a session across all trials for large- and small-reward blocks separately. To examine the dynamics of $d'$, $c$, and percent correct at $i$th trial ($i = 1–120$), we measured block-averaged values for each reward conditions. Overall $d'$ was measured by:

$$d'_{\text{overall}} = \sqrt{\left(d'^2_{\text{InRF}} + d'^2_{\text{OutRF}}\right)} \qquad (3)$$

where $d'_{\text{InRF}}$ and $d'_{\text{OutRF}}$ are the sensitivities in the two hemifields, inside and outside the recorded neurons' RFs.

*Pupil area.* All pupil area measurements were measured binocularly at 500 Hz, while monkeys maintained fixation in absence of a luminosity change using infrared camera (EyeLink 1000, SR Research). Raw pupil areas were normalized (between 0 and 1) for each session and each eye separately as:

$$\text{normalized pupil area} = \frac{\text{raw pupil area} - \text{Pupil}_{\text{Min}}}{\text{Pupil}_{\text{Max}} - \text{Pupil}_{\text{Min}}} \qquad (4)$$

where $\text{Pupil}_{\text{Max}}$ and $\text{Pupil}_{\text{Min}}$ are, respectively, the maxima and minima of raw pupil areas over 0–400 ms from sample stimulus within a session for a given eye. Normalized pupil areas of both eyes were then averaged. Mean pupil area was measured by averaging the normalized pupil area over 400 ms from sample appearance.

*Neuronal response.* Only neurons with an average spike rate 60–260 ms after sample stimulus onset that was significantly ($p < 0.01$) greater than the rate 0–250 ms before sample onset were used in the analysis. To construct PSTHs for figures, spike trains were smoothed with a half Gaussian kernel (standard deviation of 15 ms with only a rightward tail), aligned to sample stimuli onset and averaged across trials. A spike rate modulation (Fig. 3d) was measured by neuronal $d'$ as the difference in averaged z-scored spike rates (60–260 ms after sample onset) between the large- and small-reward blocks (correct trials). Spike rate modulation index (MI) for other physiological and neuronal correlates (Supplementary Table 1) was calculated as:

$$\text{MI}_X = \frac{\langle X_{\text{large}} \rangle - \langle X_{\text{small}} \rangle}{\langle X_{\text{large}} \rangle + \langle X_{\text{small}} \rangle} \qquad (5)$$

For analyzing trial-by-trial dynamics of spike counts and GLM, absolute spike counts within 60–260 ms from sample stimuli onset (or test 1 onset) were used.

PCA was done on spike rates from 0 to 400 ms separately for each monkey in Matlab. Each neuron had two spike rates, one each for one type of reward block (small or large reward). For monkey P, there were total of 407 single and multiunits (814 spike trains), and for monkey S, there were total of 563 single and multiunits (1126 spike trains). Linear regression was fit between PC scores of spike histograms in small- and large-reward blocks for the first three PC.

*Fano factor.* Mean-matched Fano factor (Fig. 3e) was measured using spike counts over 50 ms sliding windows in 2 ms steps for each neuron according to procedures mentioned previously[65]. Then the variance and mean across trial was computed at every time bin. The greatest common distribution of means across neurons, attentional intensities, and time bins was measured. In order to match the mean distribution to the common mean distribution, a different subset of neurons was randomly chosen (20 times) at every time bin and the average Fano factor was computed (ratio of the variance to the mean).

*Spike-count correlations.* Pearson correlation coefficients were computed for pair of simultaneously recorded units on spike counts over 200 ms (60–260 ms from sample stimuli onset), defined as the covariance of spike counts normalized by the variances of individual neurons:

$$\rho_{12} = \frac{\text{Cov}(r_1, r_2)}{\sqrt{\text{Var}(r_1) \times \text{Var}(r_2)}} \tag{6}$$

where $r_1$ and $r_2$ are spike counts of neuron 1 and neuron 2 across trials. Pairwise spike-count correlations were binned according to the geometric mean of the evoked responses of the two neurons in 5 Hz intervals. Evoked response was computed and subtracting by the trial-averaged baseline spike rate (−200 to 0 ms from sample onset) from the trial-averaged spike rate during the sample (60–260 ms from sample onset) (Fig. 3f). The last bin (25 Hz) included all evoked responses > 25 Hz.

*Overlap between neuron's receptive field (RF) and sample stimulus.* The strength of visual drive of a RF by the sample Gabor stimulus was estimated by the extent of overlaps between the spatial RF and the stimulus (Fig. 4). For each neuron, spatial RF was measured and fit using a bivariate Gaussian. We then calculated overlaps between probability densities of spatial RF (bivariate Gaussian fit) and the Gabor stimulus. The overlap varied from 0 to 100%, where 0 being no overlap (very weak visual drive by the stimulus) and 100% being complete overlap between the RF and the stimulus (maximum visual drive).

*Reward-matched average spike counts.* Trials were sorted into nine different bins with 50% overlap (bin size, 0.2; overlap, 0.1) depending on the normalized reward size on the preceding trial. Trials were further separated based on whether they occurred within the first 1–10 trials or within the last 60 trials (61–120) from the first correct response after the reward switch. Spike counts of each neuron were measured for these two sets of trial groups.

*Linear regression: pupil area–spike count–attentional intensity.* Average pupil area over 0–400 ms from sample onset and spike counts from 60 to 260 ms on each individual trial were used for linear regression analysis for pupil area versus spike count and average attentional intensity ("low" and "high") (Fig. 5j, left). The pupil area was modeled as:

$$\text{pupil area} = a * (\text{spike count}) + b * (\text{attentional intensity}) \tag{7}$$

where $a$ and $b$ are the regression coefficients. All the trials across small- and large-reward conditions that occurred after the first correct response (hit or CR) following the reward switch were included in the analysis. In a second model:

$$\text{pupil area} = a * (\text{spike count}) \tag{8}$$

each neuron was fit separately for two reward conditions, small and large (Fig. 5j, right). For this model, we used a subset of trials with matched pupil areas between the two reward conditions. Standardized coefficients were compared across neurons that were fit at significant level of $p < 0.05$ (*F*-test).

*Pupil area–spike count cross-correlation.* We measured cross-correlations between two time series, spike rates and pupil area, as a function of time lag over 700 ms period around the sample stimulus on (−350 to 350 ms) for each neuron. Single-trial spike trains were binned in a 10 ms sliding window (2 ms increments) and converted to spike rates. Pupil area time series was sampled at 500 Hz and directly used for the cross-correlation analysis. Single-trial cross-correlations were averaged, and trial-shuffled values were subtracted separately for trials with small- and large-reward conditions.

*Spike-triggered averaged pupil area.* We measured a STA-pupil area for each neuron within a time window of 0–400 ms from sample stimulus onset, across all trials. The time series of pupil areas were aligned to individual spikes, and averaged. Finally, trial-shuffled STA was subtracted from this averaged STA-pupil area for each neuron.

*Generalized linear model (GLM).* GLM regression was used to estimate the relationship between single-trial spike responses and reward expectancies, attentional intensity, behavior choices, and stimulus parameters. Single-trial stimulus evoked spike counts were modeled to follow a negative binomial distribution. The negative binomial distribution is well suited for the purpose, as spike-count variances of cortical neurons are most often equal to or greater than their means (Supplementary Fig. 21)[65,66]. Reward expectancy was represented by reward history on ten trials preceding the trial being considered. Mean pupil area during 400 ms following stimulus onset served as a proxy for attentional intensity. A categorical

saccade choice variable could take two values: saccade (hit and FA) or no saccade (CR and miss). For fitting of test 1 spike counts, an alternate categorical task variable "stimulus orientation change" (Δori) was used instead (Fig. 7a). This perceptual variable had two values: orientation change (nonmatch trial) or no orientation change (match trial). The product of the absolute orientation of Gabor stimulus and neuron's orientation tuning filter served as a test 1 stimulus feature variable. There were four different orientations of test 1 stimulus. There were no correlations (partial) among predictor variables (Supplementary Fig. 22). For the GLMs on test 1 spikes, in addition to the predictor variable covariances, we also cross-checked the correlation between model-predicted spike counts due to test 1 stimulus feature and Δori in order to isolate the orientation tuning-related effect on the perceptual variable Δori. We considered only the neurons that showed correlation (Spearman) of <0.2 for further analysis.

For a sample $y_1, y_2, ..., y_n$ of independent response variable $y$ from an exponential family, there exists a linear predictor $\beta_0 + \beta_1 x_1 + \cdots + \beta_k x_k$ of the response variable $y$. The mean response ($\mu_y = E[y|x_1, ..., x_k]$) depends on the linear predictor through a link function $g(\cdot)$:

$$g(\mu_y) = \beta_0 + \beta_1 x_1 + \cdots + \beta_k x_k \tag{9}$$

where $x_j$ ($j = 1, 2, ..., k$) is a set of independent predictor variables. Assuming that the spike-count responses in nonoverlapping time intervals are independent and variance is equal or greater than (over dispersion) the mean, they can be modeled as negative binomial variables. As the mean spike count is always positive, it can be modeled as:

$$\mu_y = \exp(\beta_0 + \beta_1 x_1 + \cdots + \beta_k x_k) \tag{10}$$

The probability density function for a random variable for which there are $y$ successes of Bernoulli trials until $\theta$th failure occurs follows a negative binomial distribution:

$$P(y; \mu_y, \theta) = \frac{\Gamma(y + \theta)}{\Gamma(\theta) y!} \frac{\mu_y^y \theta^\theta}{(\mu_y + \theta)^{y+\theta}} \tag{11}$$

with a mean $\mu_y$ and variance $V(y)$:

$$\mu_y = \frac{\theta p}{1 - p} \tag{12}$$

$$V_y = \mu_y + \frac{\mu_y^2}{\theta} \tag{13}$$

where $p$ is probability of success of each Bernoulli trial. $1/\theta$ is the dispersion parameter ($\theta$ is also called as shape parameter). $\Gamma$ is gamma distribution. The likelihood for a negative binomial random variable $y_i$ ($i = 1, 2, ..., n$) that depends on a set of $k$ predictors of $x_{i,j}$ ($j = 1, 2, ..., k$):

$$L(P) = \prod_{i=1}^{n} \frac{\Gamma(y_i + \theta)}{\Gamma(\theta) y_i!} \frac{\mu_{y_i}^{y_i} \theta^\theta}{\left(\mu_{y_i} + \theta\right)^{y_i + \theta}} \tag{14}$$

The log-likelihood is:

$$\text{LL}(\beta_0, \beta_1, \ldots, \beta_k, \theta) = \log(L(P)) \tag{15}$$

The predictor coefficients $\beta_j$ ($j = 1, 2, ..., k$) and shape parameter $\theta$ are obtained from maximum likelihood estimates by solving:

$$\frac{\partial \text{LL}}{\partial \beta_j} = 0, j = 1, 2, \ldots, k \tag{16}$$

$$\frac{\partial \text{LL}}{\partial \theta} = 0 \tag{17}$$

GLM was implemented in Matlab separately for each neuron. In order to compare different predictor coefficients, they were converted post hoc into standardized coefficients:

$$\beta_j' = \beta_j \times \frac{\sigma_y}{\sigma_x}, j = 1, 2, \ldots, k \tag{18}$$

Standardized coefficient corresponds to the log ratios of the mean responses (in units of standard deviation) due to one standard deviation change in the predictor variable, holding all other variables constant.

Goodness of GLM fit and model comparison. Goodness of fit for a given GLM was measured by residual deviance ($D$) and pseudo $R^2$ (Cragg and Uhler's method) value. Residual deviance measures how close the predicted values from the fitted model match the actual values from the raw data:

$$D = 2[\text{LL}(\text{saturated model}) - \text{LL}(\text{model})] \tag{19}$$

The saturated model comprises a model for which the number of parameters equals the number of samples. Thus, the predicted response in the saturated model is same as the observed response. Deviance measures lack of fit in maximum log-likelihood estimations similar to the residual variance in ordinary least squares methods. The lower the deviance, the better is the model fit. For a large sample size and small deviance, it

approximately follows chi-square distribution of $n - k - 1$ degrees of freedom. A statistic that indicates value of a proposed model is based on how well the model predicts data compared to a null model having a single predictor $\beta_0$:

$$\chi^2 = D(\text{null model}) - D(\text{model}) \qquad (20)$$

$$= 2[\text{LL}(\text{model}) - \text{LL}(\text{null model})] \qquad (21)$$

The test statistic $\chi^2$ follows an approximate chi-square distribution of degrees freedom $n - k$. We tested at a level of significance $p < 0.05$.

As the ratio of the likelihoods reflects the improvement of the proposed model over the null model (the smaller the ratio, the greater the improvement), a pseudo $R^2$ was computed based on Cragg and Uhler's approach that falls between 0 and 1:

$$R^2 = \frac{1 - \left[\frac{L(\text{null model})}{L(\text{model})}\right]^{2/n}}{1 - L(\text{null model})^{2/n}} \qquad (22)$$

where $n$ is the number of observations in the data set. Different models were compared by their $R^2$ values.

**Predictor importance.** The absolute value of the $z$-statistic of each estimated predictor coefficient measures a relative importance of that predictor variable:

$$z_j = \left| \frac{\beta_j}{\text{SE}\left(\beta_j\right)} \right|, \; j = 1, 2, \ldots, k \qquad (23)$$

where SE is the standard error.

**Cross-validation.** Predictive performance of the GLMs was measured by 10 fold $(K)$ cross-validation. Observations in each neuron's data set were split at random into $K$ partitions. GLM fit on $K - 1$ training partitions was repeated and the remaining partition was used for validation. This cross-validation was repeated $K$ times at each time one of the partitions served as the validation set:

$$\text{Error} = \frac{1}{n} \sum_{k=1}^{K} \sum_{i=1}^{n_k} \left(y_{i,k} - \hat{y}_{i,k}\right)^2 \qquad (24)$$

**Decoding of saccade selection, behavioral choice, and orientation change detection.** Saccade selection between saccade (hit and FA) and no saccade (miss and CR) was decoded from spike counts during the test 1 stimulus period based on Bayesian inference of maximum posterior probability using saccade GLM (Fig. 7d). According to Bayes' rule, the posterior probability density of the estimated choice given a spike-count response is:

$$P(x|Y) = \frac{P(Y|x)P(x)}{P(Y)} \qquad (25)$$

where $P(Y)$ is a normalization term independent of $x$. Also, assuming that there is no prior knowledge on $x$, $(P(x) = \text{constant})$, maximizing the posterior is equivalent to maximizing the log-likelihood function used in the GLM encoding model. A random pair of trials one from saccade trial (hit or FA) and another from no saccade trial (miss or CR) were selected. For each of these two trials, we evaluated the likelihoods of observed spike counts being consistent with spike counts under the two possible saccade responses (saccade versus no saccade) for each well fitted ($p < 0.05$) neuron within each session. The sum of the log-likelihood ratios across the neurons recorded in that session amounts the predicted probability of a saccade choice on that trial. The discrimination of a saccade from no saccade was counted as correct only when the decoded saccade commitment on both trials of the selected pair match with the observed data. Thus, the chance performance is 0.25, the joint probability of a set of two binary random variables.

Prediction of a choice response comprises either selecting a hit over a miss (nonmatch trials), or selecting a CR over a FA (match trials). Spike counts during the test 1 stimulus were fit with the same saccade GLM mentioned in the previous section (Figs. 6a, bottom, and 7d–f) except the miss and FA trials were randomly resampled in order to balance the number of different trial types (hits, misses, CRs, and FAs) within each session. This was to avoid any bias in decoding accuracy toward a particular choice (hits over misses or CRs over FAs), as the number of correct trials (hits and CRs) was systematically higher compared to the number of incorrect trials (misses and FAs). For measuring choice prediction, decoded saccades were converted into four choices—hit, miss, CR, and FA according to the decoded saccade values and whether the trial was a match trial or nonmatch trial. A pair of trials was randomly selected either from hit and miss (nonmatch trials) or FA and CR (match trials). A correct prediction of decoded choice required both of the selected trials to be correctly discriminated (hit from miss or FA from CR). Chance level for correctly predicting hit over miss and CR over FA is 0.25.

Similar to the saccade choices, prediction accuracy for an orientation change detection was estimated using a Δori GLM fit on the test 1 spike counts (Fig. 7a). A random pair of trials one from nonmatch trials and another from match trials was selected. Probabilities of an orientation change detection on these trials were estimated based on the sum of the log-likelihood ratios across simultaneously recorded neurons ($p < 0.05$; Δori GLM) in that session. An instance of accurate decoding of an orientation change occurred when both of the trials correctly predicted observed orientation change over a no change. The

chance level is 0.25. In Fig. 7f, individual sessions were sorted according to the number of neurons that were well fit ($p < 0.05$) with the GLM encoding model.

*Statistical analysis.* Unless otherwise specified, we used paired $t$-test and multifactor repeated measured ANOVA for comparing normally distributed data sets. Normality was checked with Kruskal–Wallis test.

**Ethical approval**. Animal experimentation: all experimental procedures were approved by the Institutional Animal Care and Use Committee protocol 72355 of the University of Chicago and were in compliance with US National Institutes of Health guidelines.

**Reporting summary**. Further information on research design is available in the Nature Research Reporting Summary linked to this article.

## Data availability
Source data are provided with this paper as an excel file (*.xlsx). All other data sets generated and analyzed in the current study are available from the corresponding author (S.G.) on reasonable request. Software codes for the behavioral task can be found in the GitHub repository (https://github.com/MaunsellLab/Lablib-Public-26-Feb-2021/tree/V1.0)[62].

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

## Acknowledgements
We thank Jackson J. Cone, Thomas Z. Luo, Zaina Zayyad, Bram-Ernst Verhoef, and Julian Day-Cooney for helpful discussion and/or comments on the paper. This work was supported by NIH grant R01EY005911. The funder had no role in study design, data collection, and interpretation, or the decision to submit the work for publication.

## Author contributions
S.G. and J.H.R.M. designed the experiments, performed the surgeries, and wrote the paper. S.G. performed the experiments and analyzed the data.

## Competing interests
The authors declare no competing interests.
