## [Peer Review File · Nature Communications]

Reviewers' Comments:

Reviewer #1:

Remarks to the Author:

Drs. Ghosh and Maunsell describe the neural correlates of attentional "intensity" in visual area V4 of the macaque brain. This intensity is dissociated from focused attention by the fact that it is "non-spatially selective". Interestingly, the neural correlates of attentional intensity and visuo-spatial attention are markedly similar (rate modulation, Fano factor, noise correlations...). This research is original because it experimentally dissociates "intensity" from "task difficulty" by modulating the reward size on a block design without taxing selective attention. Several temporal dynamics of behavior and neural activity are described as the monkeys transition from high to low intensity, and back. Single-trial encoding models dissociate the relative contribution of task and behavioral variables to spiking modulations.

The research is conducted with a solid methodology that is state-of-the-art in the visual neurophysiology field. The analyses are sound and the conclusions supported by the data. The manuscript is clear and well written. Although I am not sure how the concept of attentional "intensity" can be dissociated from other psychological constructs such as arousal, motivation, or engagement, I believe that investigating the effects of any of those constructs on neural activity is worthwhile. I do provide some suggestion below on how these constructs might be better dissociated with additional analyses.

Minor comments

I found Figure 1 to be really helpful to understand the concept that the authors are investigating.

How many neurons were recorded per session? Numbers on page 35 indicate a very low number despite using 96-channel arrays. When I divide the entire number of reported neurons by the number of sessions, I count only about 40 units per session. Why is that?

Please indicate the reaction time window limit used in the task.

Although the d-prime is a nice relative measure, it would be nice to also have an intuition of the behavioral performance in absolute terms, relative to chance... Could you please add that? For example, what is the true positive rate? What is the false alarm rate?

The use of the term "sensitivity" to represent (true positive rate – false alarm) is misleading. In statistics and epidemiology, "sensitivity" usually refers to the ratio of true positive over all positives (TP + FN). Could you think of another term? (Perhaps accuracy?)

Page 7. Pupil size – It is not clear whether the repeated ANOVA refers to a main effect of reward size, irrespective of task epoch (fix & sample), or something else... I was expecting to see two P-values, one for each epoch (simple effects). I would also expect to see some asterisks on Fig 2d.

Page 7. Pupil size correlations – Would it be possible to add the scatter plots for these three correlations in the supplementary materials?

Have you tried to correlate pupil size with reward size on a trial by trial basis rather than on a block basis? (I understand you cannot do that with d') Maybe reaction time could also be an indicator of arousal also that could be correlated on a trial-by trial basis with other variables?

If the authors have intraoperative photography of the array location relative to sulcal landmark, I would like to see those added to the supplementary material. Using arrays vs. recording chamber has this under-appreciated advantage of revealing the exact location where you are recording, and this important anatomical information should be better represented in neurophysiology papers in general,

in my opinion.

Figure 3 - I praise the authors for showing *separately* the population averages of each monkeys in b and c. Most authors would have pooled both monkeys and computed an average, which, with 95CI as error bars, would have entirely hidden the inter-monkey variability so apparent in the current figure. I do not think that this variability across monkeys (in the shape of the average PSTH) should be taken against the authors' point. Rather, I think it reveals a fascinating inter-individual variability in NHP neurophysiology that is under-appreciated by current practices, and that might be a clue into something of importance. Confirming the precise anatomical location of the arrays, as stated in my previous point, becomes even more important to interpret this variability.

The effects showed in figure 3 are convincing, albeit small. A small effect size at the neural level, however, could have a big impact on behavioral performance, since we still do not understand how modulations of this type directly impact computations at the network level. One thing that I find useful, however, is to compare one effect size with another to provide some sense of its meaning. Here is my suggestion. If the phenomenon the authors are studying is fully explained by arousal rather than attentional "intensity", then one would expect that, as arousal decreases, the observed neural modulation would decrease. Since it is a fact that, over long testing sessions, the arousal of monkeys performing behavioral tasks decrease (as they become satiated with the reward and tired), there is an opportunity for the authors to make a comparison. I would like the authors to compute the same modulation analysis they have in Figure 3 a,b,c (for reward size) by comparing trials at the beginning of the session (when arousal and motivation is high) to trials at the end of the session (when arousal is decreased). If a modulation is found, I would like the authors to compare the size of this neural modulation (caused by arousal) with the size of the modulation caused by reward size, as currently depicted. Comparing those two modulations would, in my opinion, be a very interesting observation that could strengthen the author's point. (Of course, one would want to run this analysis separately for Large and Small reward trials no to confound the effects)

I think I understand what the authors are trying to say with the analyses in Figure 4, but overall, I think the rationale could be made clearer. I think it would help the reader to state that neurons with receptive fields close but slightly off from the focus of attention are not generally modulated in selective attention.

Figure 5 a-d is very interesting... however, the e-g hysteresis plots were hard to interpret and didn't seem to add any new information not already represented in a-d. Maybe I misunderstood something.

In the decoding and encoding analyses, did the authors only consider simultaneously recorded neurons? In my opinion, single trial predictions make sense for simultaneously recorded ensembles only.

Discussion, page 29 – the authors state: "Together, these results isolate the dynamics of multiplexed representations in V4 of cognitive and experimental signals that are necessary for performance in attention demanding tasks."

The word "necessary" here carries a causal meaning. I am sure the authors do not imply that the neural correlates of non-selective attention depicted in their article imply a causal relationship to behavioral performance. Nonetheless, this sentence could be interpreted as such. Please rephrase to make it clear that no causal relationship between the neural data and the cognitive or behavioral performance are inferred from these correlational results.

Sentence starting at line 535 – "While..." Problematic syntax.

Line 591 – "remained week"

In the discussion, I would appreciate a sentence or two ruling out the possibility that the observed

effects in this research are explained by a multi-focus of spatial attention, as described in Niebergall et al. Neuron, 2011 for example.

Overall, great work.
-Sébastien Tremblay

Reviewer #3:

Remarks to the Author:

In their manuscript 'Single trial dynamics of attentional intensity in visual area V4' Ghosh and Maunsell modified their behavioral paradigm to probe the role of varying reward size in a predictable way to manipulate what they refer to as 'attentional intensity'. Given previous work from the Maunsell group the authors have a good experimental handle on fine-tuning the behavior, which allows them to manipulate attentional intensity roughly independently of selective attention, and examine the resulting effect on neuronal responses in V4. Previous work (Baruni et al. 2015) has found that neuronal responses in V4 are modulated by the expected reward size for stimuli presented inside a neuron's receptive field, in a similar way as the well characterized modulation by spatial attention. The current study also observes such modulation by the block-wise changes in available reward size, in addition to effects on variability and co-variability mirroring those of selective spatial attention. But the block-design of the current study also allows the authors to go beyond the previous findings by Baruni et al. (2015) and examine the time-course over which this neuronal modulation driven by the changes in available reward size occurs. They discover that the time-course of the neuronal modulation, the change in behavioral performance and attentional intensity as inferred from pupil size modulations are more sluggish compared to the sudden changes in available reward size. Using general linear modeling (GLM) the authors also examine the relative predictive contributions to the spiking activity of different co-variates (reward history, stimulus, pupil size, motor command, choice), and examine decoding performance for stimulus changes, saccades and choices.

The experiments and analyses appear carefully conducted, and the results are interesting. The most valuable contribution of this study in my opinion is the careful analysis of the time-courses of the different behavioral and cognitive signals (Fig. 5). Given that these findings are central and that the behavioral time-courses typically show some variability between subjects, it would be important to show these results separately for each animal. The GLM analysis is nice although some of the decoding results are puzzling (e.g. decoding accuracy for choice seems to be below chance level). Overall, I am supportive of this study but have a number of comments and concerns that the authors should be able to address with additional analyses.

Specific comments:

1) Time-courses of cognitive signals and behavior:

Fig. 5a-d: what are the time-courses in each animal individually? Which time-constants differ statistically from each other?

-Baruni et al. (2015) found a substantial effect of available reward size on the percentage of aborted trials, as expected for a change in motivation. How did the available reward size affect early aborts in the current study, and what was the time-course of that?

-Given that the early aborts or fixation breaks likely differ systematically between large and small available reward blocks—how do the time-courses compare if plotted as a function of real time rather than number of trials?

-Why is the firing rate on trial 1 for block transition from small to large not at the level of that of the end of the small block? Did the animals anticipate the block change? What do the data look like if all 120 trials/block are shown?

2) Fig. 7: Why is the choice decoding accuracy below chance level for higher neuron counts? How is the data across neurons combined for the decoding analysis?

3) Fig. 1: 'iso-intensity' lines. The concentric circles of 'iso-intensity' lines seem misleading. They indicate that the same discrimination performance for two targets (e.g. at $d'=1$) requires the same attentional intensity (or 'mental effort') as performance at $d'=1$ for only one target.

But "demand of joint performance \geq sum of separate demands" (e.g. Kahnemann, attn & effort). Shouldn't the 'iso-intensity' lines be concave?

(Note that the second statement related to the schematic, i.e. changes along radial lines imply no change in selectivity is not affected by such a change.)

4) Fig. 5i and corresponding text. To avoid confusion and in light of the substantial modulation by expected reward it should be clarified throughout that 'reward' refers to 'reward history' rather than expected reward, including in the y-axis label.

5) Fig. 3a-c: is the neuronal modulation with reward size prior to stimulus onset significant?

6) Contrast increases typically cause pupil constriction. What is the time-course of the pupil size locked on stimulus onset?

7) Line 72: 'reward size varied randomly from trial-to-trial': this is misleading as reward size was predictable within a block.

8) The link between 'arousal' and 'attentional intensity' is addressed in the discussion. To orient readers it would be helpful to acknowledge the link from the outset.

9) Table S1: 'microsaccade towards attended stimulus' – what does 'attended stimulus' refer to?

10) Recent work (Cowley et al. 2020) also reported substantial correlation between pupil size and neuronal activity in V4—this study should be referenced.

Minor:

- The perspective-plots (6e, 7a) make it hard to compare the values.
- typos: missing articles, e.g. in lines 70, 177, 178, 181, 189, 320, 516
- line 330: cite Joshi et al. 2016
- line 620 readout -> read out
- line 523 immerges -> emerges
- line 548 changes [...] is -> are
- line 553 changes [...] depends -> depend
- line 765 spik -> spike

Reviewer #4:

Remarks to the Author:

The manuscript titled "Single trial dynamics of attentional intensity in visual area V4" by Ghosh and Maunsell provides a thorough examination of the influence of effort/attentional intensity on behavioral sensitivity and neuronal firing in V4. Two previous studies showed that performing a more difficult task (higher effort) modulates the strength of spatial attention effects at the neuronal level (Boudreau et al, 2006; Spitzer et al, 1988). The task design of the current study enables the authors to isolate the contribution of what they term "attentional intensity", related to reward expectation or effort, but separating it from the spatial selection in spatially cued attention tasks. In blocks of high reward they found the behavioral sensitivity for detecting a change in orientation for a stimulus (either in or outside of the recorded RF) was increased, the neuronal firing rates were increased, variability and spike count correlations were reduced. They present a tour de force in analysis using GLM models to rule out several other possible mitigating factors including reward history and saccade choice. Of

particular interest, they found that pupil size is correlated with attentional intensity and might reflect that top-down cognitive states influence mechanisms for arousal or alertness associated with activation of the autonomic sympathetic system, which could modulate gain in visual cortex. The manuscript is well written and has already been well polished. The data and analyses that are presented are sound, and if anything may be presented in too much detail in some places (possibly in Fig 6 and Fig 7). However, I have no major concerns or issues with the manuscript other than a few places for points of clarity and discussion.

Minor comments:

Reference 8 (Heeger and Reynolds, 2009) is not related to attention and spike count correlations (normalization model of mean firing rate effects)?

Figure 3b,c: The mean firing rate in monkey P (panel b) exhibits a transient and then a sustained response out to about 300ms, followed by an offset in response. The x-axis indicates the stimulus sample had duration of 200ms. The same is true for the offset for monkey C (panel c). Given the neurons had on average a latency of 60ms, then I suppose the offset would begin at 260ms similar to how the onset begins around 60ms. But without visual reference, it appears offset begins later than 260ms? Was the duration of the stimulus always 200ms or are other sessions included with a longer stimulus? Or perhaps stimulus duration was slightly longer by a frame or two (210 or 220 ms?) why is the offset of the visual response delayed?

Lines 381-383,403 appear to be redundant with later discussion of pupil size and LC activity. Here the key finding is how pupil size acts as a predictor for results in Fig 6, not its physiological interpretation.

Figure 7e,f: Some improvement in clarify is possible in Figure 7. If I understand correctly, saccade choice distinguishes if a saccade was made into the RF, thus hit & FA trials from CR & miss trials? Choice compares (Hit & CR, "correct", from FA & miss, "incorrect", trials)? It appears from 7D as if saccade and choice are binary variables? But the chance performance for the decoder is 25% (Fig 7e,f). I'm not clear why chance performance is 25% or exactly what is then being decoded? Last, there was no discussion for why "choice" at larger numbers of neurons trends downwards in performance.

Line 591: "week" should be "weak"

Lines 601-604: In regards to discussion of saccade choice modulating V4 firing and references 17,18 (Moore lab), it should be noted that the GLM excluded any trials in which a saccade was initiated within the test 1 interval (60-260ms), so only 44% of those trials were included (line 421-422 of text). Thus these analyses may have excluded the set of faster saccades under 260ms in which the "pre-saccadic" effects might have most strongly modulated V4 activity. Some clarification may be necessary to distinguish the spikes predicting the subsequent saccade choice (go or stay) from the modulation of visual responses prior to a saccade towards the RF ("pre-saccadic attention", refs 17-18).

We thank all the reviewers for their insightful and constructive questions and comments on our manuscript. These have been very helpful in guiding our revisions, and have significantly strengthened the manuscript.

We have addressed each of the key issues identified by the reviewers. In the following sections we provide a detailed response to each point raised and describe new analyses and text that have been incorporated into the revised manuscript.

REVIEWER COMMENTS

Reviewer #1 (Remarks to the Author):

Drs. Ghosh and Maunsell describe the neural correlates of attentional “intensity” in visual area V4 of the macaque brain. This intensity is dissociated from focused attention by the fact that it is “non-spatially selective”. Interestingly, the neural correlates of attentional intensity and visuo-spatial attention are markedly similar (rate modulation, Fano factor, noise correlations...). This research is original because it experimentally dissociates “intensity” from “task difficulty” by modulating the reward size on a block design without taxing selective attention. Several temporal dynamics of behavior and neural activity are described as the monkeys transition from high to low intensity, and back. Single-trial encoding models dissociate the relative contribution of task and behavioral variables to spiking modulations.

The research is conducted with a solid methodology that is state-of-the-art in the visual neurophysiology field. The analyses are sound and the conclusions supported by the data. The manuscript is clear and well written. Although I am not sure how the concept of attentional “intensity” can be dissociated from other psychological constructs such as arousal, motivation, or engagement, I believe that investigating the effects of any of those constructs on neural activity is worthwhile. I do provide some suggestion below on how these constructs might be better dissociated with additional analyses.

Minor comments:

1. I found Figure 1 to be really helpful to understand the concept that the authors are investigating. How many neurons were recorded per session? Numbers on page 35 indicate a very low number despite using 96-channel arrays. When I divide the entire number of reported neurons by the number of sessions, I count only about 40 units per session. Why is that?

We analyzed only those units that were stable throughout recording sessions, which lasted for 3 to 5 h. That duration limited the number of units per session. Not every electrode provided useful data during each session (monkey P, 37 to 53 units/session; monkey S, 21 to 51 units/session). We have made this point clear in the “Electrophysiological Recording and Data Collection” section in “MATERIALS AND METHODS”.

2. Please indicate the reaction time window limit used in the task.

Monkeys had to respond within 500 ms (Monkey P) and 470 ms (Monkey S) after the appearance of the test stimulus. We have made changes to the “Behavioral task” section in “MATERIALS AND METHODS” to include this detail.

3. Although the d-prime is a nice relative measure, it would be nice to also have an intuition of the behavioral performance in absolute terms, relative to chance. Could you please add that? For example, what is the true positive rate? What is the false alarm rate?

We have included a new supplementary Figure S2 and supplementary Table S2 to address the reviewer’s request about the Hit and False alarm rates.

Supplementary figure S2: Mean behavioral performance across sessions. a Hit and FA rates during small and large reward blocks for monkey P (N = 9 sessions). **b** Same as in **a** for monkey S (N = 15 sessions). Error bars, ± SEM.

Supplementary table S2: Mean behavioral hit and false alarm (FA) rates

	Hit rate (%) (Mean ± SEM)		False alarm rate (%) (Mean ± SEM)	
	Small reward	Large reward	Small reward	Large reward
Monkey P (N = 9)	85.9 ± 2.2	92.8 ± 0.7	34.7 ± 2.4	13.1 ± 1.4
Monkey S (N = 15)	79.1 ± 1.4	89.1 ± 1.0	30.3 ± 1.4	15.4 ± 1.3

4. The use of the term “sensitivity” to represent (true positive rate – false alarm) is misleading. In statistics and epidemiology, “sensitivity” usually refers to the ratio of true positive over all positives (TP + FN). Could you think of another term? (Perhaps accuracy?)

In this article, we have adopted the formal definition of behavioral sensitivity, or d' , from the “Signal detection theory” model described by Macmillan and Creelman¹. Sensitivity was computed as the difference between the inverse cumulative distributions of hit and FA rates. Sensitivity measures subjects’ ability to detect a signal from a noise independent of the response bias (or criterion). We have clarified behavioral measure of sensitivity in detail in the “Behavioral performance” section of “Data Analysis” in ‘MATERIALS and METHODS’.

5. Page 7. Pupil size – It is not clear whether the repeated ANOVA refers to a main effect of reward size, irrespective of task epoch (fix & sample), or something else... I was expecting to

see two P-values, one for each epoch (simple effects). I would also expect to see some asterisks on Fig 2d.

We have included the following revision in ‘Behavioral control of attentional intensity’ section in ‘RESULTS’.

‘...(repeated measures ANOVA, effect of reward size, $F_{(1, 415)} = 45.5$, $p < 10^{-10}$; effect of sample stimulus, $F_{(1, 415)} = 180.12$, $p < 10^{-15}$; Fig. 2d)....’

6. Page 7. Pupil size correlations – Would it be possible to add the scatter plots for these three correlations in the supplementary materials? Have you tried to correlate pupil size with reward size on a trial by trial basis rather than on a block basis? (I understand you cannot do that with d') Maybe reaction time could also be an indicator of arousal also that could be correlated on a trial-by trial basis with other variables?

We have included a new supplementary Figure S4 showing the scatter plots for the correlations between behavioral sensitivity (d'), pupil area and reward size.

Single trial reward size and pupil area were very weakly correlated ($\rho = 0.006$, $p = 0.15$). We agree with the reviewer that in addition to the pupil area, the reaction time can an informative measure of animal’s attentiveness. Animals in our behavior task were not challenged to respond fast as the window used for saccade response was relatively long (monkey P, 500 ms; monkey S, 470 ms). Thus, in comparison to behavioral sensitivity (d') and percent correct, the reaction times were not a reliable measure of the subjects’ performances. Consequently, we restricted our single trial analysis to blocked averaged trial-by-trial values.

Supplementary figure S4: Correlations between block-by-block trial averaged behavioral d' , pupil area and reward size. (a-c) Distributions of mean reward size versus behavioral d' (a); reward size versus pupil area (b) and behavioral d' versus pupil area (c) across all reward-blocks (N = 417).

7. If the authors have intraoperative photography of the array location relative to sulcal landmark, I would like to see those added to the supplementary material. Using arrays vs. recording chamber has this under-appreciated advantage of revealing the exact location where

you are recording, and this important anatomical information should be better represented in neurophysiology papers in general, in my opinion.

We have included a new supplementary Figure S5, supplementary Table S3 and revised text in 'METHODS and MATERIALS' to address the reviewer's comment about the electrode placements.

There was a small difference between the electrode array placements. The array was placed more dorsal with respect to the inferior occipital sulcus (ios) in monkey S compared to monkey P. Polar angles of the recorded units' receptive fields (RF) differed in the two monkeys (mean \pm SEM, monkey P, $132.4^\circ \pm 2.6^\circ$; monkey S, $108.8^\circ \pm 3.2^\circ$; $p < 10^{-11}$, ranksum). Eccentricities of RFs in the two animals did not differ (mean \pm SEM eccentricity, monkey P, $2.8^\circ \pm 0.1^\circ$; monkey S, $2.7^\circ \pm 0.1^\circ$; $p = 0.18$, ranksum). Similarly, RF sizes between animals were not different (mean \pm SEM RF size (1 sigma), monkey P, $1.1^\circ \pm 0.05^\circ$; monkey S, $1.2^\circ \pm 0.06^\circ$; $p = 0.76$, ranksum).

Supplementary figure S5: Electrode array placements. a-b Reconstructed brain structural MRI of monkey P (a) and monkey S (b). sts, Superior temporal sulcus; ls, lunate sulcus; ios, inferior occipital sulcus. Green box with grids, 10x10 electrode array. c-d Spatial receptive field (RF) centers of all unique units (only one unit from each electrode contact) from monkey P (c) and

monkey S (d). e-g Distributions of RF eccentricities (e), polar angles (f) and RF size (sigma; g) for both monkeys. There is no difference between the eccentricities ($p = 0.18$, ranksum test; e) and RF sizes ($p = 0.76$, ranksum test; g) of recorded units in the two monkeys. RF Polar angles between the two animals differed significantly ($p < 10^{-11}$, ranksum test; f).

Supplementary table S3: Spatial RF location and size

	RF eccentricity (°) Mean \pm SEM	RF polar angle(°) Mean \pm SEM	RF size, sigma (°) Mean \pm SEM
Monkey P (n = 66)	2.8 \pm 0.1	132.4 \pm 2.6	1.13 \pm 0.05
Monkey S (n = 81)	2.7 \pm 0.1	108.8 \pm 3.2	1.18 \pm 0.06

8. Figure 3 - I praise the authors for showing *separately* the population averages of each monkey in b and c. Most authors would have pooled both monkeys and computed an average, which, with 95CI as error bars, would have entirely hidden the inter-monkey variability so apparent in the current figure. I do not think that this variability across monkeys (in the shape of the average PSTH) should be taken against the authors' point. Rather, I think it reveals a fascinating inter-individual variability in NHP neurophysiology that is under-appreciated by current practices, and that might be a clue into something of importance. Confirming the precise anatomical location of the arrays, as stated in my previous point, becomes even more important to interpret this variability.

We appreciate the reviewer's comment. We have described in anatomical locations of recording arrays and recorded neurons' receptive field properties in previous response to Comment #7.

9. The effects showed in figure 3 are convincing, albeit small. A small effect size at the neural level, however, could have a big impact on behavioral performance, since we still do not understand how modulations of this type directly impact computations at the network level. One thing that I find useful, however, is to compare one effect size with another to provide some sense of its meaning. Here is my suggestion. If the phenomenon the authors are studying is fully explained by arousal rather than attentional "intensity", then one would expect that, as arousal decreases, the observed neural modulation would decrease. Since it is a fact that, over long testing sessions, the arousal of monkeys performing behavioral tasks decrease (as they become satiated with the reward and tired), there is an opportunity for the authors to make a comparison. I would like the authors to compute the same modulation analysis they have in Figure 3 a,b,c (for reward size) by comparing trials at the beginning of the session (when arousal and motivation is high) to trials at the end of the session (when arousal is decreased). If a modulation is found, I would like the authors to compare the size of this neural modulation (caused by arousal) with the size of the modulation caused by reward size, as currently depicted. Comparing those two modulations would, in my opinion, be a very interesting observation that could strengthen the author's point. (Of course, one would want to run this analysis separately for Large and Small reward trials no to confound the effects).

As the reviewer points out, the animal's motivation (due to change in satiation) and arousal (due to fatigue) decrease with trial progression within a session. Thus, it will be informative to

compare the effects of attentional intensity and collective change in a generic level of arousal and motivation on behavioral performance, physiology and neuronal responses. To address this, we separately analyzed behavioral d' , rate of aborted trials (fixation breaks), pupil area and V4 spike counts for the early (first half) and late (last half) trials within each session. We have included these results in supplementary Figure S9, supplementary Table S4 and text in the **'Responses of V4 neurons increase with increasing attentional intensity'** section in 'RESULTS', which reads as follows:

"It is expected that with increased reward size, animals' level of general arousal might also increase. Next, we wanted to test if the observed effects due to changes in attentional intensity can be fully explained by general arousal. It is believed that animal's general arousal and motivation decreases with time during a long experimental session which can be reflected by pupil size and the rate of aborted trials². We separately analyzed behavioral d' , rate of aborted trials (fixation breaks), pupil area and V4 spike counts for the early (first half) and late (last half) trials for small and large rewards within each session (**Supplementary Fig. S9; Supplementary Table S4**). We then compared the modulations of these measures between attentional intensity and session-timing. Attentional intensity significantly affected behavioral d' ($p < 10^{-3}$) and the rate of aborted trials ($p < 10^{-3}$), whereas trial timing (early versus late) had no significant effects on behavioral d' ($p > 0.05$) or aborted trials ($p > 0.05$). Thus, the level of attentiveness relevant to the task in a given type of reward block (small versus large) did not change detectably over longer intervals within a session irrespective of changes in satiation (motivation) and fatigue (or general arousal). In contrast, pupil area and V4 spike counts were significantly affected by both the attentional intensity ($p < 10^{-3}$) and trial timing ($p < 10^{-4}$). However, there were no significant effects of intensity-by-trial timing interaction on pupil area or spike responses ($p > 0.05$). Together, these results suggest that the attentional intensity and general arousal can independently modulate common downstream targets such as autonomic sympathetic nervous system (pupil dilation) and cortical neuronal activity."

Supplementary figure S9: Comparing behavioral (d'), physiological (pupil area) and neuronal modulations between early and late halves of trials within a session. Trials were divided into two halves, early trials and late trials in every session for each monkey. **a, e** Behavioral sensitivity (d') across blocks during early and late trials with reward changes (#sessions, $N = 9$ for monkey P; $N = 15$ for monkey S). Excluding the reward size (small versus large), there was no significant effect of within session trial time (early versus late) on behavioral d' (monkey P, reward size, $p < 10^{-4}$, $F_{(1, 32)} = 100.94$; trial time, $p = 0.09$, $F_{(1, 32)} = 2.96$; reward-by-trial time interaction, $p = 0.4$, $F_{(1, 32)} = 0.7$; two way ANOVA). **b, f** Mean rate of aborted trials for monkey P and monkey S. **c, g** Mean pupil area during the sample stimulus period ((#blocks, $N_{\text{small}} = 77$, $N_{\text{large}} = 86$, monkey P; $N_{\text{small}} = 128$, $N_{\text{large}} = 126$, monkey S). **d, h** PSTHs of V4 neuronal spike rates (#units, $n = 407$, monkey P; $n = 563$, monkey S). Error bars, \pm SEM.

Supplementary table S4: Modulation of behavioral, physiological and neurophysiological responses during early and late trials within a session.

	Monkey P					
	Reward size	Early trials	Late trials	ANOVA		
				Attention intensity	Trial timing	Intensity-by-time interaction
Behavioral d'	Small	1.51 ± 0.13	1.59 ± 0.11	$F_{(1,32)} = 100.94$ ($p < 10^{-4}$)	$F_{(1,32)} = 2.96$ ($p = 0.09$)	$F_{(1,32)} = 0.7$ ($p = 0.4$)
	Large	2.45 ± 0.08	2.72 ± 0.08			
	Small	46.9 ± 3.5	56.3 ± 4.2			

%Abort trials	Large	31.9 ± 1.9	33.9 ± 1.9	F(1,32) = 36.9 (p < 10 ⁻⁴)	F(1,32) = 3.41 (p = 0.07)	F(1,32) = 1.43 (p = 0.24)
Pupil area	Small	1.043 ± 0.007	0.947 ± 0.011	F(1,159) = 108.86 (p < 10 ⁻⁴)	F(1,159) = 160.26 (p < 10 ⁻⁴)	F(1,159) = 1.77 (p = 0.18)
	Large	1.143 ± 0.014	1.027 ± 0.008			
V4 spike rate	Small	0.807 ± 0.008	0.764 ± 0.010	F(1,16267) = 67.84 (p < 10 ⁻¹⁵)	F(1,16267) = 67.44 (p < 10 ⁻¹⁵)	F(1,16267) = 3.01 (p = 0.08)
	Large	0.927 ± 0.006	0.843 ± 0.009			
Monkey S						
	Reward size	Early trials	Late trials	ANOVA		
				Attention intensity	Trial timing	Intensity-by-time interaction
Behavioral d'	Small	1.47 ± 0.07	1.25 ± 0.07	F(1,56) = 162.6 (p < 10 ⁻⁴)	F(1,56) = 1.38 (p = 0.24)	F(1,56) = 3.55 (p = 0.06)
	Large	2.26 ± 0.07	2.31 ± 0.07			
%Abort trials	Small	54.4 ± 2.2	58.5 ± 1.8	F(1,56) = 7.56 (p = 0.008)	F(1,56) = 0.05 (p = 0.83)	F(1,56) = 4.12 (p = 0.05)
	Large	53.2 ± 1.7	49.9 ± 1.3			
Pupil area	Small	1.004 ± 0.005	0.956 ± 0.005	F(1,250) = 32.85 (p < 10 ⁻⁴)	F(1,250) = 96.4 (p < 10 ⁻⁴)	F(1,250) = 0.11 (p = 0.74)
	Large	1.038 ± 0.006	0.984 ± 0.005			
V4 spike rate	Small	0.851 ± 0.007	0.825 ± 0.006	F(1,22516) = 16.08 (p < 10 ⁻⁴)	F(1,22516) = 21.34 (p < 10 ⁻⁴)	F(1,22516) = 0.5 (p = 0.48)
	Large	0.892 ± 0.007	0.886 ± 0.006			

10. I think I understand what the authors are trying to say with the analyses in Figure 4, but overall, I think the rationale could be made clearer. I think it would help the reader to state that neurons with receptive fields close but slightly off from the focus of attention are not generally modulated in selective attention.

We have included the following revision in the ‘**Spatial distribution of neuronal modulation**’ section in results in the revised manuscript:

“Responses of V4 neurons with RF away from the focus of spatially selective attention are generally not modulated. Thus, an absence of spike modulation for the neurons with RFs slightly off from the focus of attention would indicate a significant contribution of spatially selective attention in the observed attention intensity related neuronal effects.”

11. Figure 5 a-d is very interesting... however, the e-g hysteresis plots were hard to interpret and didn't seem to add any new information not already represented in a-d. Maybe I misunderstood something.

We have included the following the revision in the '**Trial-by-trial behavioral, physiological and neuronal dynamics in response to reward modulation**' section in the Results:

"The discrepancies between the effects of two directions of reward changes on neuronal firing, pupil area and behavior can be clearly seen in hysteresis plots (**Fig. 5e-g**). These response transitions show memory effects where an equivalent change in the response requires significantly different amount of reward change depending on the change direction."

12. In the decoding and encoding analyses, did the authors only consider simultaneously recorded neurons? In my opinion, single trial predictions make sense for simultaneously recorded ensembles only.

We considered only simultaneously recorded neurons in a session for decoding analysis. We have clarified this point in the '**Dynamic representation of the detection of behaviorally relevant sensory signal in area V4**' section in RESULTS.

"Sum of the log-likelihood ratios across **simultaneously recorded** neurons amounts to the predicted probability of a realization of orientation-change on that trial."

13. Discussion, page 29 – the authors state: "Together, these results isolate the dynamics of multiplexed representations in V4 of cognitive and experimental signals that are necessary for performance in attention demanding tasks." The word "necessary" here carries a causal meaning. I am sure the authors do not imply that the neural correlates of non-selective attention depicted in their article imply a causal relationship to behavioral performance. Nonetheless, this sentence could be interpreted as such. Please rephrase to make it clear that no causal relationship between the neural data and the cognitive or behavioral performance are inferred from these correlational results.

We apologize for the lack of clarity. We have revised the following text in the 'DISCUSSION' section.

"Together, these results isolate the dynamics of multiplexed representations of cognitive and experimental signals in V4 which can be crucial for performance in attention demanding tasks."

14. Sentence starting at line 535 – "While..." Problematic syntax.

We have corrected the relevant text in the 'DISCUSSION' section.

15. Line 591 – "remained week"

We have corrected the text in the 'DISCUSSION' section.

16. In the discussion, I would appreciate a sentence or two ruling out the possibility that the observed effects in this research are explained by a multi-focus of spatial attention, as described in Niebergall et al. Neuron, 2011 for example.

We have added the following text in the 'DISCUSSION' section in the revised manuscript:

"Previous studies have reported that visual spatial attention can be split into multiple foci to suppress task irrelevant distractor information^{3,4}. Absence of any distractor in the current task and spatially non selective neuronal modulation with changes in attention intensity eliminates the possibility of multi foci split spatial attention mediated effects in the results."

Reviewer #3 (Remarks to the Author):

In their manuscript 'Single trial dynamics of attentional intensity in visual area V4' Ghosh and Maunsell modified their behavioral paradigm to probe the role of varying reward size in a predictable way to manipulate what they refer to as 'attentional intensity'. Given previous work from the Maunsell group the authors have a good experimental handle on fine-tuning the behavior, which allows them to manipulate attentional intensity roughly independently of selective attention, and examine the resulting effect on neuronal responses in V4. Previous work (Baruni et al. 2015) has found that neuronal responses in V4 are modulated by the expected reward size for stimuli presented inside a neuron's receptive field, in a similar way as the well characterized modulation by spatial attention. The current study also observes such modulation by the block-wise changes in available reward size, in addition to effects on variability and co-variability mirroring those of selective spatial attention. But the block-design of the current study also allows the authors to go beyond the previous findings by Baruni et al. (2015) and examine the time-course over which this neuronal modulation driven by the changes in available reward size occurs. They discover that the time-course of the neuronal modulation, the change in behavioral performance and attentional intensity as inferred from pupil size modulations are more sluggish compared to the sudden changes in available reward size. Using general linear modeling (GLM) the authors also examine the relative predictive contributions to the spiking activity of different co-variates (reward history, stimulus, pupil size, motor command, choice), and examine decoding performance for stimulus changes, saccades and choices.

The experiments and analyses appear carefully conducted, and the results are interesting. The most valuable contribution of this study in my opinion is the careful analysis of the time-courses of the different behavioral and cognitive signals (Fig. 5). Given that these findings are central and that the behavioral time-courses typically show some variability between subjects, it would be important to show these results separately for each animal. The GLM analysis is nice although some of the decoding results are puzzling (e.g. decoding accuracy for choice seems to be below chance level). Overall, I am supportive of this study but have a number of comments and concerns that the authors should be able to address with additional analyses.

Specific comments:

1. Time-courses of cognitive signals and behavior:

1a. Fig. 5a-d: What are the time-courses in each animal individually? Which time-constants differ statistically from each other?

To address the reviewer's comment, we added a new supplementary Figure S13 and supplementary Table S5 where trial-by-trial dynamics of behavioral d' , pupil area and V4 spike counts are separated for individual animals.

Supplementary figure S13: Block averaged trial-by-trial dynamics of behavioral sensitivity (d'), pupil area and V4 neuronal spiking with reward changes for monkey P (a, c, e and g; #blocks, $N_{\text{small}} = 77$, $N_{\text{large}} = 86$) and monkey S (b, d, f and h; #blocks, $N_{\text{small}} = 128$, $N_{\text{large}} = 126$). Circles, observed data. Lines, single exponential fits. τ , decay or rise constants. Trials are aligned with respect to the first correct trial following a block transition. Dashed lines, 95% confidence intervals. a, b Received rewards. c, d Behavioral sensitivity (d'). e, f Normalized mean pupil area

during sample stimulus period. **g, h** Normalized V4 spike counts across all recorded neurons (monkey P, n = 407; monkey S, n = 563).

Supplementary table S5: Single-trial decay/rise constant (τ) for behavior (sensitivity, d'), physiology and V4 neurophysiology in response to reward changes.

	Behavior ($\tau_{d'}$, 95% CI)		Pupil area (τ_{pupil} , 95% CI)		V4 spiking (τ_{neuron} , 95% CI)	
	small \rightarrow large	large \rightarrow small	small \rightarrow large	large \rightarrow small	small \rightarrow large	large \rightarrow small
Monkey P ($N_{small} = 77$; $N_{Large} = 86$)	11.4 (5.6, 17.3)	1.1 (-0.1, 2.3)	2.7 (2.2, 3.1)	13.7 (12.4, 15.0)	2.6 (1.5, 3.7)	11.1 (8.4, 13.8)
Monkey S ($N_{small} = 128$; $N_{Large} = 126$)	15.6 (8.5, 22.6)	0.8 (-0.1, 1.7)	5.1 (4.1, 6.0)	8.7 (6.9, 10.5)	1.4 (-0.4, 3.1)	6.3 (3.4, 9.1)

1b. Baruni et al. (2015) found a substantial effect of available reward size on the percentage of aborted trials, as expected for a change in motivation. How did the available reward size affect early aborts in the current study, and what was the time-course of that?

To address the reviewer’s comment, we added a new supplementary Figure S14 and included the following text in the ‘**Trial-by-trial behavioral, physiological and neuronal dynamics in response to reward modulation**’ section in ‘RESULTS’:

“Similar to previous studies², we also found that the changes in reward size differentially affected trial-by-trial rate of aborted trials depending on the direction of reward change (**Supplementary Fig. S14**). Immediately after the block transition, the aborted trials slowly increased ($\tau = 4.2$) for transitions from large to small compared transitions from small to large ($\tau = 0.1$). However, as the trials progressed abort trial rate gradually decreased in small reward blocks.”

Supplementary figure S14: Trial-by trial aborted trials. Block averaged trial-by-trial aborted trials (fixation breaks) for large (N = 212) and small (N = 205) reward conditions. Circles, observed data. Lines, single exponential fits. τ , decay or rise constants. Trials are aligned with respect to the first correct trial following block transition. Dashed lines, 95% confidence intervals.

1c. Given that the early aborts or fixation breaks likely differ systematically between large and small available reward blocks—how do the time-courses compare if plotted as a function of real time rather than number of trials?

To address the reviewer’s comment, we converted the trial-by-trial dynamics of behavioral d' , pupil area and V4 spike counts as a function of absolute timing of the trials. We have included the following text in the ‘**Trial-by-trial behavioral, physiological and neuronal dynamics in response to reward modulation**’ section in ‘RESULTS’ and a new supplementary Figure S15.

“Taking into account the dissimilar early aborted trial rates between small and large reward blocks did not alter qualitatively trial-by-trial dynamics of behavioral d' , pupil area and V4 spike counts as a function of time rather than trial count (**Supplementary Fig. S15**). Time constants of these behavioral, physiological and neurophysiological variables were closely proportional to the trial constants as measured from trial-by-trial dynamics as a function of number of trials.”

Supplementary figure S15: Block averaged temporal dynamics of behavioral sensitivity (d'), pupil area and V4 neuronal spiking with reward changes as a function of time (#blocks, small, 205; large, 212; two monkeys). **a** Rewards received. **b** Behavioral sensitivity (d'). **c** Mean pupil area during sample stimulus. **d** Mean normalized spike counts across blocks and neurons ($n=970$). τ , decay or rise constants in seconds. Trials are aligned with respect to the first correct trial following block transition. Dashed lines, 95% confidence intervals.

1d. Why is the firing rate on trial 1 for block transition from small to large not at the level of that of the end of the small block? Did the animals anticipate the block change? What do the data look like if all 120 trials/block are shown?

Blocks were aligned with the first correct response (hit or CR) after the block transition, which was the first time the animal could have known that a transition had occurred. Thus, the block averaged spike count on the trial 1 for small to large transition reward block comes entirely from correct trials which is not the case for the final trials in small reward blocks. Consequently, the average spike count on the 1st trial in large reward blocks was higher compared to the final trial in small reward blocks.

Because of the block alignment with the first correct trial, several blocks contained fewer than 120 trials. For consistency in displaying results, we plotted up to 100 trials. The dynamics and results did not differ when 120 trials were considered.

2. Fig. 7: Why is the choice decoding accuracy below chance level for higher neuron counts? How is the data across neurons combined for the decoding analysis?

Individual sessions were sorted according to the number of neurons that were well fit ($p < 0.05$) with the GLM encoding model in Figure 7f. For a given trial, log-likelihood ratios of all well fit neurons recorded in that session were combined to estimate the probability of a saccade over no-saccade. According to the saccade probability every trial's decoded response was converted into hit, miss, CR and FA. A pair of trials were randomly selected from match trials (or non-match trials), one hit (or FA) and one miss (or CR) trials. A prediction was considered correct when the decoded responses of this pair of trials correctly discriminated a hit from miss response (or a CR from FA). Thus, the chance performance is 0.25, the joint probability of a set of two binary random variables. The log-likelihood ratio and the probability of detecting a saccade over no-saccade increases with the number of neurons used for the decoding within a session (Figure 7f). Prediction accuracy for choice can become less than chance (0.25) if there is a bias for correctly decoding a particular trial type over another trial type. We did not see any such bias on saccade decoding accuracy, but there was a bias towards higher decoding accuracy for Hits and CRs compared to misses and FAs in many sessions.

We clarified this in the '**Decoding of saccade selection, response choice and orientation change detection**' section in 'METHODS and MATERIALS'.

3. Fig. 1: 'iso-intensity' lines. The concentric circles of 'iso-intensity' lines seem misleading. They indicate that the same discrimination performance for two targets (e.g. at $d' = 1$) requires the same attentional intensity (or 'mental effort') as performance at $d' = 1$ for only one target. But "demand of joint performance \geq sum of separate demands" (e.g. Kahnemann, attn & effort). Shouldn't the 'iso-intensity' lines be concave? (Note that the second statement related to the schematic, i.e. changes along radial lines imply no change in selectivity is not affected by such a change.)

We agree with the reviewer that the joint "mental effort" for maintaining same level of performance for two targets as according to subjective reports can be higher compared to an individual target⁵. Thus, it would have been more appropriate for the absolute cognitive demand or mental effort in a joint target task to follow a concave like profile. In this manuscript, we restricted the use of the term "attentional intensity" as an objective function of behavioral d 's

because a true measure of mental effort is difficult. It is very likely that attentional intensity and mental effort might have a monotonic relationship, but that has not been explored in detail in this study.

We have included the following text revision in the ‘INTRODUCTION’ section associated with Figure 1.

“Schematically, attentional intensity is represented here as an objective function of resultant performance across targets. It may not fully capture all aspects of perceived subjective mental effort⁵.”

4. Fig. 5i and corresponding text. To avoid confusion and in light of the substantial modulation by expected reward it should be clarified throughout that ‘reward’ refers to ‘reward history’ rather than expected reward, including in the y-axis label.

We have revised axis label in Figure 5i and text in the ‘Trial-by-trial behavioral, physiological and neuronal dynamics in response to reward modulation’ section in ‘RESULTS’.

5. Fig. 3a-c: is the neuronal modulation with reward size prior to stimulus onset significant?

We have reported the results on the modulation of V4 spike counts during the pre-sample fixation period in supplementary Figure S6 and included text revision in the ‘RESULTS’ section.

“Trial averaged population spike rates during the during the pre-sample fixation period for correctly completed trials (hits and correct rejections) did not differ significantly between the two reward trials ($p = 0.06$; **Supplementary Fig. S6**).”

6. Contrast increases typically cause pupil constriction. What is the time-course of the pupil size locked on stimulus onset?

We have included the following supplementary Figure S3 on the time course of pupil area locked to sample stimulus onset. There was a constriction once the animals fixated and before the sample onset. Pupil area peaked >250 ms after the sample onset.

Supplementary figure S3: Session averaged stimulus-evoked pupil area for small and large rewards. Pupil areas on single trials were first aligned with the sample stimulus onset. Time course of mean pupil area within a session was z-scored with respect to the pre-sample fixation period (–400 to 0 ms) separately for two different reward size. Left, monkey P (N = 9). Right, monkey S (N = 15). Error bars, \pm SEM.

7. Line 72: ‘reward size varied randomly from trial-to-trial’: this is misleading as reward size was predictable within a block.

Reward size within a block was predictably large or small. However, relative rewards for a hit and correct rejection responses were varied within blocks to encourage the animal to maintain a behavioral criterion near zero. We have included the following text revision in the ‘INTRODUCTION’ section to clarify this.

“Additionally, the relative reward size for the hit and correct rejections were varied within blocks in an uncued way in order to encourage the animals to maintain a behavioral criterion near zero. This variance in reward size also allowed us to produce single trial estimates of the dynamics of neuronal correlates of reward expectation as well as attentional intensity.”

8. The link between ‘arousal’ and ‘attentional intensity’ is addressed in the discussion. To orient readers it would be helpful to acknowledge the link from the outset.

We have included the following text revision in the ‘INTRODUCTION’ section.

“Attention and arousal are thought to be closely related. An individual is expected to be aroused in order to maintain a high level of performance while engaged in a demanding task. It is plausible that the intensive component of attention is a specific type of more general arousal, but it remains unknown how this top-down intensity signal is represented by cortical neuronal activity.”

9. Table S1: ‘microsaccade towards attended stimulus’ – what does ‘attended stimulus’ refer to?

Attended stimulus refers to the two stimulus locations in the opposite hemifields. We have clarified this in supplementary Table S1.

10. Recent work (Cowley et al. 2020) also reported substantial correlation between pupil size and neuronal activity in V4—this study should be referenced.

We agree with the reviewer that the research article by Cowley et al. (2020) is interesting and relevant. We have included this reference in the ‘DISCUSSION’ section of the revised manuscript.

“A more recent study has shown that V4 spiking activity strongly covaried fluctuations in pupil size association with spontaneous slow drift in task performance⁶.”

Minor Comments:

11. The perspective-plots (6e, 7a) make it hard to compare the values.

We have revised the two plots and added grids to make comparisons easier.

12. typos: missing articles, e.g. in lines 70, 177, 178, 181, 189, 320, 516

We have corrected the typos in the relevant sections of the revised manuscript.

13. line 330: cite Joshi et al. 2016

We have included this citation in the **'Trial-by-trial behavioral, physiological and neuronal dynamics in response to reward modulation'** section in 'RESULTS'.

14. line 620 readout -> read out

We have corrected the typo in 'DISCUSSION' section.

15. line 523 immerges -> emerges

We have corrected the typo in 'DISCUSSION' section.

16. line 548 changes [...] is -> are

We have corrected the typo in 'DISCUSSION' section.

17. line 553 changes [...] depends -> depend

We have corrected the typo in 'DISCUSSION' section.

18. line 765 spik -> spike

We have corrected the typo in 'Data analysis' section in "MATERIALS and METHODS.

Reviewer #4 (Remarks to the Author):

The manuscript titled "Single trial dynamics of attentional intensity in visual area V4" by Ghosh and Maunsell provides a thorough examination of the influence of effort/attentional intensity on behavioral sensitivity and neuronal firing in V4. Two previous studies showed that performing a more difficult task (higher effort) modulates the strength of spatial attention effects at the neuronal level (Boudreau et al, 2006; Spitzer et al, 1988). The task design of the current study enables the authors to isolate the contribution of what they term "attentional intensity", related to reward expectation or effort, but separating it from the spatial selection in spatially cued attention tasks. In blocks of high reward they found the behavioral sensitivity for detecting a change in orientation for a stimulus (either in or outside of the recorded RF) was increased, the neuronal firing rates were increased, variability and spike count correlations were reduced. They present a tour de force in analysis using GLM models to rule out several other possible mitigating factors including reward history and saccade choice. Of particular

interest, they found that pupil size is correlated with attentional intensity and might reflect that top-down cognitive states influence mechanisms for arousal or alertness associated with activation of the autonomic sympathetic system, which could modulate gain in visual cortex. The manuscript is well written and has already been well polished. The data and analyses that are presented are sound, and if anything may be presented in too much detail in some places (possibly in Fig 6 and Fig 7). However, I have no major concerns or issues with the manuscript other than a few places for points of clarity and discussion.

Minor comments:

1. Reference 8 (Heeger and Reynolds, 2009) is not related to attention and spike count correlations (normalization model of mean firing rate effects)?

We have corrected this with the following reference in the 'REFERENCE' section.

"8 Mitchell, J. F., Sundberg, K. A. & Reynolds, J. H. Spatial attention decorrelates intrinsic activity fluctuations in macaque area V4. *Neuron* **63**, 879-888 (2009)."

2. Figure 3b,c: The mean firing rate in monkey P (panel b) exhibits a transient and then a sustained response out to about 300ms, followed by an offset in response. The x-axis indicates the stimulus sample had duration of 200ms. The same is true for the offset for monkey C (panel c). Given the neurons had on average a latency of 60ms, then I suppose the offset would begin at 260ms similar to how the onset begins around 60ms. But without visual reference, it appears offset begins later than 260ms? Was the duration of the stimulus always 200ms or are other sessions included with a longer stimulus? Or perhaps stimulus duration was slightly longer by a frame or two (210 or 220 ms?) why is the offset of the visual response delayed?

Duration of the sample stimulus was always fixed, 200 ms for both animals. We agree that the offset response latency was higher than the onset response. A more recent study suggests that the offset response can be ~10-20 ms longer than the onset response and it varies with the type of behavioral task⁷. A similar long offset latency in V4 and IT neurons was found in several previous studies⁸⁻¹⁰. We have included the following text revision in the 'Responses of V4 neurons increase with increasing attentional intensity' section in 'RESULTS'.

"The offset response latency was longer than the onset latency. A similar offset latency difference in V4 and IT neurons was found in several previous studies⁸⁻¹⁰. A strong top-down modulation and activation of recurrent network could increase the offset latency in V4 neurons depending on the type of behavioral task⁷."

3. Lines 381-383,403 appear to be redundant with later discussion of pupil size and LC activity. Here the key finding is how pupil size acts as a predictor for results in Fig 6, not its physiological interpretation.

We have edited the relevant text in the revised manuscript.

4. Figure 7e,f: Some improvement in clarify is possible in Figure 7. If I understand correctly, saccade choice distinguishes if a saccade was made into the RF, thus hit & FA trials from CR & miss trials? Choice compares (Hit & CR, “correct”, from FA & miss, “incorrect”, trials)? It appears from 7D as if saccade and choice are binary variables? But the chance performance for the decoder is 25% (Fig 7e,f). I’m not clear why chance performance is 25% or exactly what is then being decoded? Last, there was no discussion for why “choice” at larger numbers of neurons trends downwards in performance.

We have clarified this point in the ‘*Generalized linear model (GLM)*’ section in ‘MATERIALS and METHODS’.

“A random pair of trials one from saccade trial (hit or FA) and another from no-saccade trial (miss or CR) were selected. For each of these two trials, we evaluated the likelihoods of observed spike counts being consistent with spike counts under the two possible saccade responses (saccade versus no-saccade) for each well fitted ($p < 0.05$) neuron within each session. The sum of the log-likelihood ratios across the neurons recorded in that session amounts to the predicted probability of a saccade choice on that trial. The discrimination of a saccade from no-saccade was counted as correct only when the decoded saccade commitment on both trials of the selected pair match with the observed data. Thus, chance performance is 0.25, the joint probability of a set of two binary random variables. For measuring choice prediction, decoded saccades were converted into four choices – hit, miss, CR and FA according to the decoded saccade values and whether the trial was a match trial or non-match trial. A pair of trials were randomly selected, either hit and miss (nonmatch trials) or FA and CR (match trials). A correct prediction of decoded choice required both of the selected trials to be correctly discriminated (hit from miss or FA from CR). Similarly, prediction accuracy for orientation change detection was estimated using Δ ori-GLM. Chance level was again 0.25. In Figure 7f, individual sessions were sorted according to the number neurons that were well fit ($p < 0.05$) with the GLM encoding model. Although, the chance level for two random variables is 0.25, choice prediction accuracy can become less than the chance level irrespective of the number of neurons used for decoding if there is a bias for correctly decoding a particular response type over the other response type.”

5. Line 591: “week” should be “weak”

We have corrected the text.

6. Lines 601-604: In regards to discussion of saccade choice modulating V4 firing and references 17,18 (Moore lab), it should be noted that the GLM excluded any trials in which a saccade was initiated within the test 1 interval (60-260ms), so only 44% of those trials were included (line 421-422 of text). Thus these analyses may have excluded the set of faster saccades under 260ms in which the “pre-saccadic” effects might have most strongly modulated V4 activity. Some clarification may be necessary to distinguish the spikes predicting the subsequent saccade choice (go or stay) from the modulation of visual responses prior to a saccade towards the RF (“pre-saccadic attention”, refs 17-18).

To address the reviewer’s comment, we have included the following text revision in the ‘DISCUSSION’ section.

“The lack of a slow latency pre-saccadic signal in V4 spikes based on the spike counts over the 200 ms stimulus period does not fully rule out the possibility of fast pre-saccadic modulation that is shorter than 200 ms.”

References

- 1 Macmillan, N. A. & Creelman, C. D. *Detection theory: A user's guide*. 208 (Psychology press, 2004).
- 2 Baruni, J. K., Lau, B. & Salzman, C. D. Reward expectation differentially modulates attentional behavior and activity in visual area V4. *Nat. Neurosci.* **18**, 1656 (2015).
- 3 Niebergall, R., Khayat, P. S., Treue, S. & Martinez-Trujillo, J. C. Multifocal attention filters targets from distracters within and beyond primate MT neurons' receptive field boundaries. *Neuron* **72**, 1067-1079 (2011).
- 4 Reardon, K. M., Kelly, J. G. & Matthews, N. Bilateral attentional advantage on elementary visual tasks. *Vis. Res.* **49**, 691-701 (2009).
- 5 Kahneman, D. *Attention and effort*. Vol. 1063 (Prentice-Hall, 1973).
- 6 Cowley, B. R. *et al.* Slow drift of neural activity as a signature of impulsivity in macaque visual and prefrontal cortex. *bioRxiv* (2020).
- 7 Zamarashkina, P., Popovkina, D. V. & Pasupathy, A. Timing of response onset and offset in macaque V4: stimulus and task dependence. *J. Neurophysiol.* **123**, 2311-2325 (2020).
- 8 Maunsell, J. H. & Gibson, J. R. Visual response latencies in striate cortex of the macaque monkey. *J. Neurophysiol.* **68**, 1332-1344 (1992).
- 9 Luo, T. Z. & Maunsell, J. H. Neuronal modulations in visual cortex are associated with only one of multiple components of attention. *Neuron* **86**, 1182-1188 (2015).
- 10 Matsumora, T., Koida, K. & Komatsu, H. Relationship between color discrimination and neural responses in the inferior temporal cortex of the monkey. *J. Neurophysiol.* **100**, 3361-3374 (2008).

Reviewers' Comments:

Reviewer #1:

Remarks to the Author:

The authors have done a great job at answering all of my questions and comments.

I would suggest a different color for the array placement in Supp. Fig. S5 instead of dark green (maybe bright red?) It was quite hard to see on my screen.

I have no further comments. Great work.

-Sébastien Tremblay

Reviewer #3:

Remarks to the Author:

The authors made commendable and detailed revisions and addressed most of my comments.

My only remaining concern is my previous point 2.

2) If a decoder performs below chance it suggests that the model is overfitting the data. The fact that this trend gets worse with the number of neurons included further suggests overfitting. This should be avoided and is typically straightforward to remedy by appropriate regularization, e.g. Lasso regularization (Tibshirani, 1996) using the L1 penalization (e.g. `lasso` in Matlab). This should be corrected.

Reviewer #4:

Remarks to the Author:

The authors have addressed the few concerns that I had from my previous review.

Response to Reviewers' Comments

Reviewer #1 (Remarks to the Author):

“The authors have done a great job at answering all of my questions and comments. I would suggest a different color for the array placement in Supp. Fig. S5 instead of dark green (maybe bright red?) It was quite hard to see on my screen. I have no further comments. Great work.”

-Sébastien Tremblay

We are thankful to **Reviewer 1** Dr. Tremblay for acknowledging that analyses and revisions made according to his comments in the revised manuscript was satisfactory.

We have revised the Supplementary Figure 5 to incorporate his suggestion for better readability.

Reviewer #3 (Remarks to the Author):

“The authors made commendable and detailed revisions and addressed most of my comments. My only remaining concern is my previous point.

2. If a decoder performs below chance it suggests that the model is overfitting the data. The fact that this trend gets worse with the number of neurons included further suggests overfitting. This should be avoided and is typically straightforward to remedy by appropriate regularization, e.g. Lasso regularization (Tibshirani, 1996) using the L1 penalization (e.g. lasso in Matlab). This should be corrected.”

We are thankful to the **Reviewer 3** for the helpful suggestion to rectify the suboptimal decoding performance of choices in Figure 7e-f. We have corrected this issue with an additional analysis and included the results in Figure 7e-f in the ‘Results’ and ‘Methods’ sections in the revised manuscript.

We identified that the decoding accuracy for both hit and correct-rejection (CR) trials were better compared to miss and false alarm (FA) trials. This systemic bias in decoding accuracy occurred as a result of unbalanced trial numbers of different choices. On an average the number of hits (or CRs) were ~2-3 fold of the numbers of misses (or FAs) trials within an experimental session. When an encoding model (Figure 7e, 7f) was used to isolate choice information (hits over misses, or CRs over FAs) on a dataset with disproportionate representations of hits over misses (and CRs over FAs), the decoding accuracy for an observed hit choice (or CR) was higher compared to a miss choice (or FA) resulting to suboptimal decoding accuracy. We have rectified this issue by matching the trial numbers of hits and misses (as well CRs and FAs) using random resampling methods. The revised analysis eliminated any bias in choice decoding accuracy and monotonically increased with the number of units used for decoding.

Reviewer #4 (Remarks to the Author):

“The authors have addressed the few concerns that I had from my previous review.”

We are grateful to **Reviewer 4** for acknowledging that the revisions in response to his/her comments have been to his/her satisfaction in the revised manuscript.